# Direct Preference-based Policy Optimization without Reward Modeling

**Gaon An**[*]
Seoul National University
white0234@mllab.snu.ac.kr

**Junhyeok Lee**[*]
Seoul National University
riman314@mllab.snu.ac.kr

**Xingdong Zuo**
NAVER
xingdong.zuo@navercorp.com

**Norio Kosaka**
NAVER
Line Corporation
kosaka.norio@linecorp.com

**Kyung-Min Kim**
NAVER
kyungmin.kim.ml@navercorp.com

**Hyun Oh Song**[†]
Seoul National University
hyunoh@mllab.snu.ac.kr

## Abstract

Preference-based reinforcement learning (PbRL) is an approach that enables RL agents to learn from preference, which is particularly useful when formulating a reward function is challenging. Existing PbRL methods generally involve a two-step procedure: they first learn a reward model based on given preference data and then employ off-the-shelf reinforcement learning algorithms using the learned reward model. However, obtaining an accurate reward model solely from preference information, especially when the preference is from human teachers, can be difficult. Instead, we propose a PbRL algorithm that directly learns from preference without requiring any reward modeling. To achieve this, we adopt a contrastive learning framework to design a novel policy scoring metric that assigns a high score to policies that align with the given preferences. We apply our algorithm to offline RL tasks with actual human preference labels and show that our algorithm outperforms or is on par with the existing PbRL methods. Notably, on high-dimensional control tasks, our algorithm surpasses offline RL methods that learn with ground-truth reward information. Finally, we show that our algorithm can be successfully applied to fine-tune large language models.

## 1 Introduction

Deep reinforcement learning has been successful in solving various decision-making tasks where a well-defined reward function is available [34, 35, 50, 5, 54]. However, in many real-world tasks, it can be challenging to design a quantitative reward function that accurately reflects the desired behavior, particularly when the task involves human evaluation. Preference-based RL (PbRL) seeks to provide an alternative solution by enabling agents to learn from preference information between pairs of trajectory segments [1, 10]. PbRL has gained considerable interest in recent years as making a relative judgment is much easier than providing a real-valued score, which makes human feedback much more viable [39, 38, 52, 45, 56].

---

[*]First two authors have equal contributions

[†]Corresponding author

37th Conference on Neural Information Processing Systems (NeurIPS 2023).

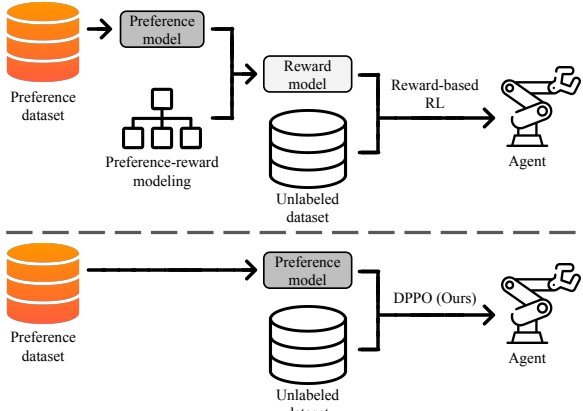

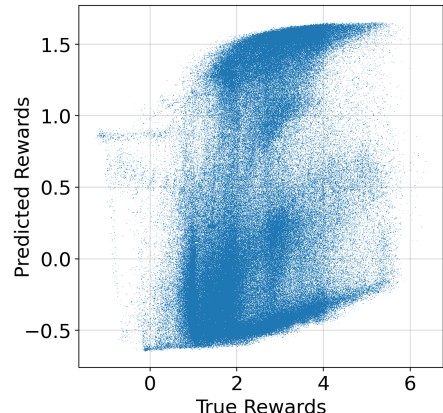

Figure 1: An overview of the difference between our approach (below) and the baselines (top). Our approach does not require modeling the reward from the preference predictor as our policy optimization algorithm can learn directly from preference labels.

Figure 2: Predicted reward vs. true reward on the Hopper environment when using a reward model from PbRL [27]. The reward model fails to accurately capture the underlying reward structure.

Recent PbRL methods take a two-step approach: they first learn a reward model from the given preference dataset and then run off-the-shelf reinforcement learning algorithms on top of the learned reward model [10, 31, 40]. However, acquiring an accurate reward model only from preference labels, typically provided by human teachers, poses a significant challenge as it is unclear how to extract the underlying reward structure from preference. Current methods rely on modeling the reward with certain specific assumptions, though there are some concerns regarding whether those assumptions hold in practice [13, 27].

Alternatively, predicting the preference itself is comparatively more straightforward since we have direct access to training labels, allowing us to leverage powerful techniques from supervised learning. Building upon this observation, we introduce a PbRL algorithm that bypasses the need for reward function modeling by directly learning from preference labels. Our approach begins by devising a policy scoring metric that assigns high scores to policies aligning with the provided preference dataset. Concretely, the PbRL objective is formulated as a contrastive learning problem, guiding the learned policy to be closer to more preferred trajectory segments while distancing itself from the less preferred ones [9, 20]. Furthermore, we enhance the performance of the preference predictors from previous works by introducing a novel prediction smoothness regularizer. Experiment results on offline RL settings with actual human preference labels show that the proposed algorithm outperforms or is on par with the baselines on all of the tasks considered [16]. Notably, in high-dimensional control tasks, our algorithm outperforms offline RL methods that utilize ground-truth reward information. Moreover, our preliminary experiments show that our algorithm can be successfully applied for fine-tuning large language models. Our official code is available at `https://github.com/snu-mllab/DPPO`.

## 2 Preliminaries

### 2.1 Preference-based reinforcement learning

Reinforcement learning considers an environment formulated as a Markov Decision Process (MDP) defined by a tuple $(\mathcal{S}, \mathcal{A}, T, \mathcal{R}, p_0, H)$, where $\mathcal{S}$ is a state space, $\mathcal{A}$ is an action space, $T(\mathbf{s}'|\mathbf{s}, \mathbf{a})$ is the state transition dynamics, $\mathcal{R}(\mathbf{s}, \mathbf{a})$ is the reward function, $p_0(\mathbf{s})$ is the initial state distribution, and $H$ is the time horizon. The goal of reinforcement learning is to learn a policy $\pi$ that optimizes the expected return:

$$J(\pi) = \mathbb{E}_{\mathbf{s}_0 \sim p_0, \mathbf{a}_t \sim \pi(\cdot|\mathbf{s}_t), \mathbf{s}_{t+1} \sim T(\cdot|\mathbf{s}_t, \mathbf{a}_t)} \left[ \sum_{t=0}^{H} r_t \right].$$

Conventional RL assumes the reward information $(r_t)$ is given and uses this to optimize their policy. However, finding a suitable reward metric can be costly in many real-world scenarios.

For example, if the goal task is to train a robot to scramble an egg, it would be unclear how to design a reward function that captures all the desired properties. Instead, PbRL assumes the supervision is given in the form of preference. Concretely, for a pair of trajectory segments $(\sigma^0, \sigma^1)$, where $\sigma^i = (\mathbf{s}_0^i, \mathbf{a}_0^i, \mathbf{s}_1^i, \mathbf{a}_1^i, \ldots, \mathbf{s}_k^i, \mathbf{a}_k^i)$, a (human) teacher indicates which segment it prefers. The preference label $y$ is given as 0 if $\sigma^0$ is preferred, *i.e.*, $\sigma^0 \succ \sigma^1$. In a similar manner, $y$ is set to 1 if $\sigma^1$ is preferred and 0.5 if the two are equally preferred.

To learn a reward model $\widehat{r}$, prior works assume the preference depends on the value of the underlying rewards summed over each timestep [10, 6, 31, 40]:

$$\widehat{P}[\sigma^0 \succ \sigma^1; \psi] = \frac{\exp\left(\sum_{t=0}^{k} \widehat{r}\left(\mathbf{s}_t^0, \mathbf{a}_t^0; \psi\right)\right)}{\exp\left(\sum_{t=0}^{k} \widehat{r}\left(\mathbf{s}_t^0, \mathbf{a}_t^0; \psi\right)\right) + \exp\left(\sum_{t=0}^{k} \widehat{r}\left(\mathbf{s}_t^1, \mathbf{a}_t^1; \psi\right)\right)},$$

where $\psi$ denotes the learnable parameters of the reward model. Given a dataset $\mathcal{D}_{\text{pref}}$ of preference triples $(\sigma^0, \sigma^1, y)$, the reward model is trained by minimizing the cross-entropy loss between the preference predictions and the ground-truth labels:

$$\ell_{\widehat{r}}(\psi; \mathcal{D}_{\text{pref}}) = - \mathop{\mathbb{E}}_{(\sigma^0, \sigma^1, y) \sim \mathcal{D}_{\text{pref}}} \left[ (1-y) \log \widehat{P}\left[\sigma^0 \succ \sigma^1; \psi\right] + y \log \widehat{P}\left[\sigma^1 \succ \sigma^0; \psi\right] \right]. \quad (1)$$

After training, any standard RL algorithm can be used to maximize the expected return under the learned reward model. Especially in the offline PbRL setting, we assume there exists a small dataset $\mathcal{D}_{pref}$ with preference labels along with a much larger unlabeled dataset $\mathcal{D}$ without any reward or preference labels [27, 49]. A typical approach involves utilizing $\mathcal{D}_{pref}$ to learn the reward model and applying the model to label $\mathcal{D}$.

## 2.2 Contrastive learning

The goal of contrastive learning is to learn representations where similar sample pairs are close to each other while dissimilar pairs are far apart. For an anchor sample $\mathbf{x}$ (*e.g.*, an image), suppose we have a positive sample $\mathbf{x}^+$ (*e.g.*, the same image with data augmentation applied) and a set of negative samples $\{\mathbf{x}_i^-\}_{i=1}^m$ (*e.g.*, samples from other images). To learn an encoder $f$, contrastive learning minimizes the following loss:

$$\ell_f\left(\mathbf{x}, \mathbf{x}^+, \{\mathbf{x}_i^-\}_{i=1}^m\right) = -\log \frac{\exp\left(f\left(\mathbf{x}\right)^\mathsf{T} f\left(\mathbf{x}^+\right)\right)}{\exp\left(f\left(\mathbf{x}\right)^\mathsf{T} f\left(\mathbf{x}^+\right)\right) + \sum_{i=1}^m \exp\left(f\left(\mathbf{x}\right)^\mathsf{T} f\left(\mathbf{x}_i^-\right)\right)},$$

where a dot product of two representations (*e.g.*, $f(\mathbf{x})^\mathsf{T} f(\mathbf{x}^+)$) is considered as the similarity score between the two samples.

## 3 Learning directly from preference

While prior PbRL approaches adopt a two-step procedure involving the construction of a reward model using the preference data, the fidelity of the learned reward model in reflecting the original reward function remains uncertain. The main challenge lies in extracting the underlying rewards from the given preference. Previous works assume that the preference for a trajectory segment can be represented as an average of the underlying rewards [10, 21]. However, considering PbRL typically assumes human teachers provide the preference labels, it is unclear whether human preferences can also be modeled in this manner. For instance, humans may focus on a specific subset of the segment while ignoring other parts [24, 27]. Empirically, we find that the reward models learned from preferences often fail to accurately capture the underlying reward structure, as illustrated in Figure 2. This issue is problematic since within the current PBRL framework, the quality of the learned policy relies heavily on the quality of the learned rewards.

Meanwhile, predicting the preference itself is a more straightforward task as we have explicit labels to train with and can therefore apply powerful tools from supervised learning [18, 11, 36]. Based on this observation, our goal is to introduce a PbRL algorithm that directly learns with preference information without the need for reward modeling, as illustrated in Figure 1. To achieve this, we design a policy scoring metric that yields a high score when a policy aligns with the given preference dataset. In other words, we assume that a desirable policy should be closer to $\sigma^0$ than $\sigma^1$ if $\sigma^0 \succ \sigma^1$. To formulate this property into an optimizable objective, we first define the *closeness* between a policy and a trajectory segment.

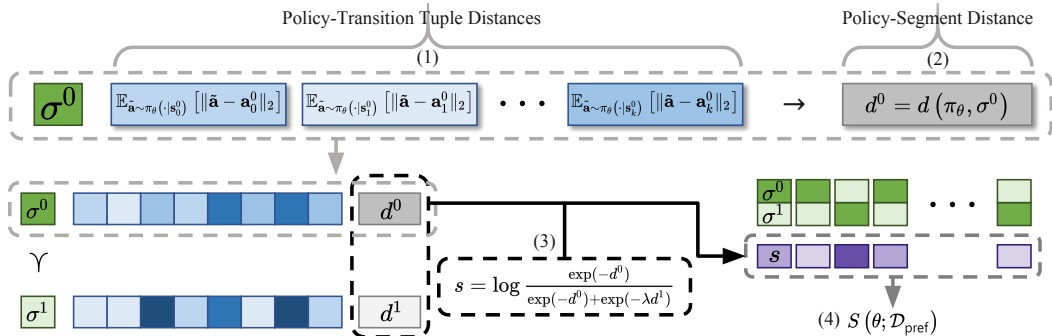

Figure 3: An overview of the score calculation process. To score a given policy, (1) the first step is to calculate the distance between each transition tuple and the policy. (2) Second, these distances are aggregated to a policy-segment distance through a predefined aggregation function. (3) Finally, we obtain the score value by contrasting the policy-segment distances according to their preference.

## 3.1 Policy-segment distance

We define the distance between a policy and a trajectory segment as an aggregation of the distance between a policy and each transition tuple in the trajectory segment. Concretely, the policy-segment distance can be expressed as

$$d\left(\pi, \sigma^i\right) = \text{AGG}\left(d_{\mathbf{sa}}\left(\pi, \mathbf{s}_0^i, \mathbf{a}_0^i\right), \ldots, d_{\mathbf{sa}}\left(\pi, \mathbf{s}_k^i, \mathbf{a}_k^i\right)\right),$$

where $d_{\mathbf{sa}}$ denotes the policy-transition tuple distance and AGG denotes an aggregation function. There can be multiple ways for instantiating $d_{\mathbf{sa}}$. For simplicity, we employ the expected $\ell_2$ distance between the policy action and the trajectory action: $d_{\mathbf{sa}}(\pi, \mathbf{s}, \mathbf{a}) = \mathbb{E}_{\tilde{\mathbf{a}} \sim \pi(\cdot|\mathbf{s})}\left[\|\tilde{\mathbf{a}} - \mathbf{a}\|_2\right]$. Similarly, we opt to use the mean operator as the aggregation function. To sum up, our policy-segment distance function becomes

$$d(\pi, \sigma^i) = \frac{1}{k+1} \sum_{t=0}^{k} \left(\mathbb{E}_{\tilde{\mathbf{a}} \sim \pi(\cdot|\mathbf{s}_t^i)}\left[\left\|\tilde{\mathbf{a}} - \mathbf{a}_t^i\right\|_2\right]\right).$$

## 3.2 Preference score metric

Using the policy-segment distance defined above, we can now build a score metric that satisfies our desired property. Given a preference triple $(\sigma^0, \sigma^1, y)$, assume that $\sigma^0$ is preferred over $\sigma^1$, *i.e.*, $y = 0$. We want to assign a high score if a policy is closer to $\sigma^0$ than $\sigma^1$. This condition can be expressed in terms of the policy-segment distance as $\sigma^0 \succ \sigma^1 \Rightarrow d(\pi, \sigma^0) < d(\pi, \sigma^1)$. To capture this condition across multiple segment pairs into a single metric, we adopt a contrastive learning formulation:

$$S(\theta; \mathcal{D}_{\text{pref}}) = \mathbb{E}_{(\sigma^0, \sigma^1, y) \sim \mathcal{D}_{\text{pref}}} \left[(1 - y) \cdot s\left(\pi_\theta, \sigma^0, \sigma^1\right) + y \cdot s\left(\pi_\theta, \sigma^1, \sigma^0\right)\right] \quad (2)$$

$$\text{s.t.} \quad s\left(\pi, \sigma^i, \sigma^j\right) = \log \frac{\exp\left(-d\left(\pi, \sigma^i\right)\right)}{\exp\left(-d\left(\pi, \sigma^i\right)\right) + \exp\left(-d\left(\pi, \sigma^j\right)\right)},$$

where $\theta$ denotes the learnable parameters of the policy $\pi$. In terms of contrastive learning, we can interpret $\exp\left(-d\left(\pi_\theta, \sigma^i\right)\right)$ as representing the similarity between the policy $\pi_\theta$ and the segment $\sigma^i$. Then, $\pi_\theta$, $\sigma^0$, and $\sigma^1$ are each considered the anchor, the positive sample, and the negative sample, assuming $\sigma^0 \succ \sigma^1$. Note that a high score can only be achieved if the policy is closer to more preferred trajectory segments.

While Equation (2) is sufficient to express our desired property, a minor drawback is that the score function is indifferent to the increase or decrease of the distances in the same magnitude. To understand this in detail, let us first denote $d^i = d(\pi_\theta, \sigma^i)$ for brevity. Then, for a preference triple $(\sigma^0, \sigma^1, y)$ with $y = 0$,

$$s\left(\pi_\theta, \sigma^0, \sigma^1\right) = -d^0 - \log\left(\exp\left(-d^0\right) + \exp\left(-d^1\right)\right) \approx \max\left\{0, d^0 - d^1\right\}. \quad (3)$$

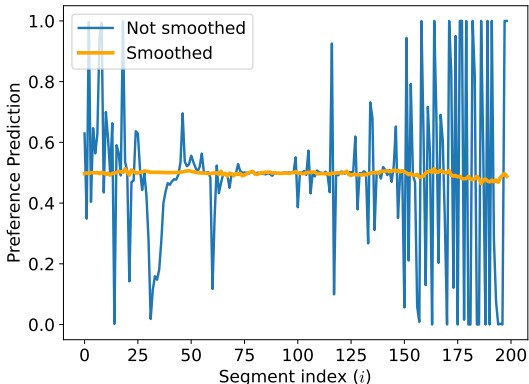

Figure 4: Predicted preference of overlapping segments from a single trajectory. In detail, we measure $\widehat{P}[\sigma^i \succ \sigma^{i+1}]$, where $\sigma^i = (\mathbf{s}_i, \mathbf{a}_i, \ldots \mathbf{s}_{i+k}, \mathbf{a}_{i+k})$.

**Algorithm 1** Direct Preference-based Policy Optimization

---

**Input:** Unlabeled dataset $\mathcal{D}$, preference dataset $\mathcal{D}_{\mathrm{pref}}$, learning rate $\eta_\phi$ and $\eta_\theta$, number of training steps $M$ and $N$, and regularization parameters $\lambda, \nu$.
Initialize network parameters $\phi$ and $\theta$
**for** step $= 1$ **to** $M$ **do**
    Update the predictor parameter:
    $\phi \leftarrow \phi - \eta_\phi \nabla_\phi \ell_{\widehat{P}}(\phi; \mathcal{D}_{\mathrm{pref}}, \mathcal{D})$
**end for**
**for** step $= 1$ **to** $N$ **do**
    Update the policy parameter:
    $\theta \leftarrow \theta + \eta_\theta \nabla_\theta S(\theta; \mathcal{D}, \phi, \lambda)$
**end for**

---

Therefore, the score value remains the same when the distances increase or decrease in the same magnitude. In other words, the score for the distances $(d^0, d^1)$ and $(d^0 + \alpha, d^1 + \alpha)$ are identical, indicating there is no penalty when a policy deviates from even the preferred trajectory segment. To solve this, we add a regularizing factor $\lambda \in (0, 1)$ that decreases the score when the overall scale of the policy-segment distances increases:

$$S(\theta; \mathcal{D}_{\mathrm{pref}}, \lambda) = \mathop{\mathbb{E}}_{(\sigma^0, \sigma^1, y) \sim \mathcal{D}_{\mathrm{pref}}} \left[ (1 - y) \cdot s\left(\pi_\theta, \sigma^0, \sigma^1; \lambda\right) + y \cdot s\left(\pi_\theta, \sigma^1, \sigma^0; \lambda\right) \right] \qquad (4)$$

$$\text{s.t.} \quad s\left(\pi, \sigma^i, \sigma^j; \lambda\right) = \log \frac{\exp\left(-d\left(\pi, \sigma^i\right)\right)}{\exp\left(-d\left(\pi, \sigma^i\right)\right) + \exp\left(-\lambda d\left(\pi, \sigma^j\right)\right)}.$$

If we set $\lambda$ smaller than 1 and plug in this new formulation to Equation (3), it is easy to find out that the score will decrease when the overall distances increase. The resulting score calculation process for Equation (4) is illustrated in Figure 3.

### 3.3 Policy optimization with preference predictor

We can directly optimize a policy with preference labels by maximizing the score function in Equation (4). However, to leverage the unlabeled dataset $\mathcal{D}$, we train a preference predictor using the labeled dataset $\mathcal{D}_{\mathrm{pref}}$. We formulate this process as a simple binary classification problem and use the cross-entropy loss to optimize the predictor $\widehat{P}$:

$$\ell_{\widehat{P}}(\phi; \mathcal{D}_{\mathrm{pref}}, \mathcal{D}) = -\underbrace{\mathop{\mathbb{E}}_{(\sigma^0, \sigma^1, y) \sim \mathcal{D}_{\mathrm{pref}}} \left[ (1 - y) \log \widehat{P}\left[\sigma^0 \succ \sigma^1; \phi\right] + y \log \widehat{P}\left[\sigma^1 \succ \sigma^0; \phi\right] \right]}_{\text{Preference Correctness}} \qquad (5)$$

$$+ \nu \underbrace{\mathop{\mathbb{E}}_{(\sigma, \sigma') \sim \mathcal{D}} \left[ \left( \widehat{P}[\sigma \succ \sigma'; \phi] - 0.5 \right)^2 \right]}_{\text{Preference Smoothness}},$$

where $\phi$ denotes the learnable parameters of the preference predictor $\widehat{P}$. The first term resembles Equation (1), but the difference is that here we directly model the preference predictor instead of modeling the reward function. The second term is a smoothness regularizer which guides the predictor to have a similar preference against two largely overlapping segments. Concretely, given $\sigma = (\mathbf{s}_i, \mathbf{a}_i, \ldots, \mathbf{s}_{i+k}, \mathbf{a}_{i+k})$, we sample $\sigma'$ from the same trajectory as $(\mathbf{s}_{i+\alpha}, \mathbf{a}_{i+\alpha}, \ldots, \mathbf{s}_{i+\alpha+k}, \mathbf{a}_{i+\alpha+k})$, where $\alpha \sim \mathcal{N}(0, m^2)$ with $m \ll k$. We observe that if no smoothness regularization is applied ($\nu = 0$), the preference can vary significantly between two almost identical segments, as illustrated in Figure 4. This behavior is undesirable as a human teacher will unlikely be able to even identify the difference between the two segments.

Figure 5: DPPO vs. BC for two example trajectory segments on the walker2d-medium-replay dataset of D4RL Gym. DPPO successfully avoids the behavior from the unpreferred trajectory segment while BC also clones the unpreferred behavior.

After training the preference predictor with Equation (5), we can train our policy with the unlabeled dataset $\mathcal{D}$ by sampling pairs of trajectory segments and labeling their preferences:

$$S(\theta; \mathcal{D}, \phi, \lambda) = \mathop{\mathbb{E}}_{(\sigma^0, \sigma^1) \sim \mathcal{D}} \left[ (1 - \widehat{y}) \cdot s \left( \pi_\theta, \sigma^0, \sigma^1; \lambda \right) + \widehat{y} \cdot s \left( \pi_\theta, \sigma^1, \sigma^0; \lambda \right) \right],$$

$$\text{s.t.} \quad \widehat{y} = \mathbb{1} \left\{ \widehat{P} \left[ \sigma^0 \succ \sigma^1; \phi \right] > 0.5 \right\}.$$

Algorithm 1 summarizes the full process of our PbRL algorithm. Figure 5 shows an example of two policies each learned using our algorithm and behavior cloning (BC), which shows that vanilla BC suffers from imitating the behavior of unpreferred trajectory segments while our algorithm successfully distances from it. We name our preference-based policy optimization algorithm as DPPO, an abbreviation for *Direct Preference-based Policy Optimization*. The sections below evaluate the performance of DPPO in the offline setting.

## 4 Experiments

### 4.1 Offline RL experiment details

Following recent PbRL works, we evaluate our algorithm on the offline setting which assumes a large unlabeled dataset $\mathcal{D}$ is given along with a much smaller preference-labeled dataset $\mathcal{D}_{\text{pref}}$ [49, 27]. We evaluate our algorithm on D4RL, a standard benchmark for offline RL, with preference datasets generated by actual human teachers [16]. We did not experiment on Antmaze, a widely used task in D4RL, due to a crucial bug in its environment implementation (please refer to Appendix F for more details). Below is a brief description of the tasks considered in our experiments:

**Gym** D4RL Gym provides datasets for the Gym locomotion environments where the goal task is to move forward while maintaining body balance. D4RL Gym contains various types of datasets each from a different data collection process. Among those datasets, we focus on the *medium-replay* and *medium-expert* datasets, which contain trajectories with the most diverse quality.

**Adroit pen** Adroit tasks involve controlling a 24-DoF robotic hand to perform tasks such as grabbing a pen or hammering a nail [44]. Adroit tests if offline RL methods can learn from human demonstrations on these high-dimensional tasks. We focus on the *pen* task since standard offline RL algorithms fail to achieve reasonable performance on other tasks, even with the reward information.

**Kitchen** Kitchen requires learning a 9-DoF robot manipulation task [19]. Similar to Adroit, the reward function is sparse and the dataset contains human demonstrations. However, Kitchen requires solving multiple sub-tasks sequentially (*e.g.*, opening the microwave, then turning off the switch) in a single episode. While Kitchen has three types of datasets (*complete*, *partial*, and *mixed*), we consider the latter two as *complete* only contains successful trajectories with almost identical quality.

For the Gym hopper, Gym walker2d, and Adroit pen tasks, we utilize publicly available human preference datasets released by [27]. For the other tasks, we generate a new preference dataset as there are no preference datasets available. To collect the human preference datasets for each task, we strictly adhere to the protocol outlined by [27]. This involves defining a set of desirable behaviors and instructing the human teacher to label preferences accordingly. For example, a desired behavior in the Hopper environment would be to maintain body balance and prevent falling down. The size of the resulting preference datasets ranges from 100 to 500 samples, depending on the specific task. For more details regarding the dataset generation process, please refer to Appendix A.

We consider PreferenceTransformer (PT) as our baseline method, which is a state-of-the-art approach in offline PbRL [27]. PT employs a transformer network to train the reward model and leverages IQL, an offline RL algorithm, for policy optimization using the learned rewards [28]. This original version is denoted as PT+IQL. Considering that the choice of policy optimization algorithm significantly impacts the performance of reward modeling methods, we also explore using CQL for policy optimization [30]. This modified version of PT is denoted as PT+CQL. Additionally, we present the performance of CQL and IQL when utilizing the ground-truth reward information for reference. Note that these reward-based RL baselines do not provide a fair comparison with our method as they have access to a much denser supervision signal. For more implementation details regarding the baselines and our algorithm, please refer to Appendix B. For more experiments including behavior-cloning baselines, please refer to Appendix C.

## 4.2 Evaluation results

Table 1: Normalized average return on D4RL Gym tasks, averaged over 5 seeds. $\pm$ denotes the standard deviation.

| | Learning with task rewards | | Learning with preference only | | |
|---|---|---|---|---|---|
| Task Name | CQL | IQL | PT+CQL | PT+IQL | DPPO (Ours) |
| halfcheetah-medium-replay | $45.7 \pm 0.6$ | $44.3 \pm 0.7$ | $27.1 \pm 17.7$ | $\mathbf{42.3 \pm 0.5}$ | $40.8 \pm 0.4$ |
| hopper-medium-replay | $84.1 \pm 14.2$ | $100.5 \pm 1.4$ | $49.1 \pm 22.0$ | $59.7 \pm 25.8$ | $\mathbf{73.2 \pm 4.7}$ |
| walker-medium-replay | $80.0 \pm 3.4$ | $74.8 \pm 3.4$ | $\mathbf{52.8 \pm 7.2}$ | $43.3 \pm 39.8$ | $50.9 \pm 5.1$ |
| halfcheetah-medium-expert | $88.5 \pm 9.7$ | $85.2 \pm 7.4$ | $77.1 \pm 0.9$ | $83.6 \pm 3.8$ | $\mathbf{92.6 \pm 0.7}$ |
| hopper-medium-expert | $103.7 \pm 7.5$ | $84.1 \pm 24.1$ | $89.2 \pm 14.4$ | $67.8 \pm 32.3$ | $\mathbf{107.2 \pm 5.2}$ |
| walker2d-medium-expert | $108.4 \pm 0.3$ | $107.5 \pm 4.4$ | $77.7 \pm 1.2$ | $\mathbf{109.8 \pm 0.4}$ | $108.6 \pm 0.1$ |
| Average | 85.1 | 82.7 | 62.2 | 67.8 | **78.8** |

Table 1 shows the evaluation results for the D4RL Gym tasks. DPPO demonstrates superior or comparable performance to the preference-based learning methods across all considered tasks. In terms of average performance, our method outperforms the baselines by a large margin with a minimum of %11p and reaches a performance level similar to the methods that learn with ground-truth rewards. Also, DPPO exhibits significantly lower variance in performance compared to the baseline methods like PT+IQL, which suffer from pronounced fluctuations in performance.

Table 2: Normalized average return on D4RL Adroit pen and Kitchen tasks, averaged over 5 seeds. $\pm$ denotes the standard deviation.

| | Learning with task rewards | | Learning with preference only | | |
|---|---|---|---|---|---|
| Task Name | CQL | IQL | PT+CQL | PT+IQL | DPPO (Ours) |
| pen-human | $44.2 \pm 7.8$ | $53.8 \pm 36.9$ | $31.6 \pm 3.3$ | $53.0 \pm 31.7$ | $\mathbf{76.3 \pm 14.4}$ |
| pen-cloned | $42.4 \pm 5.1$ | $51.3 \pm 37.1$ | $18.3 \pm 10.6$ | $42.9 \pm 24.4$ | $\mathbf{75.1 \pm 7.7}$ |
| Average | 43.3 | 52.6 | 25.0 | 48.0 | **75.7** |
| kitchen-mixed | $10.7 \pm 10.8$ | $50.6 \pm 6.2$ | $12.3 \pm 7.7$ | $48.0 \pm 11.9$ | $\mathbf{52.5 \pm 3.1}$ |
| kitchen-partial | $12.9 \pm 13.0$ | $58.8 \pm 6.5$ | $14.1 \pm 13.0$ | $40.2 \pm 12.3$ | $\mathbf{49.4 \pm 5.7}$ |
| Average | 11.8 | 54.7 | 13.2 | 44.1 | **51.0** |

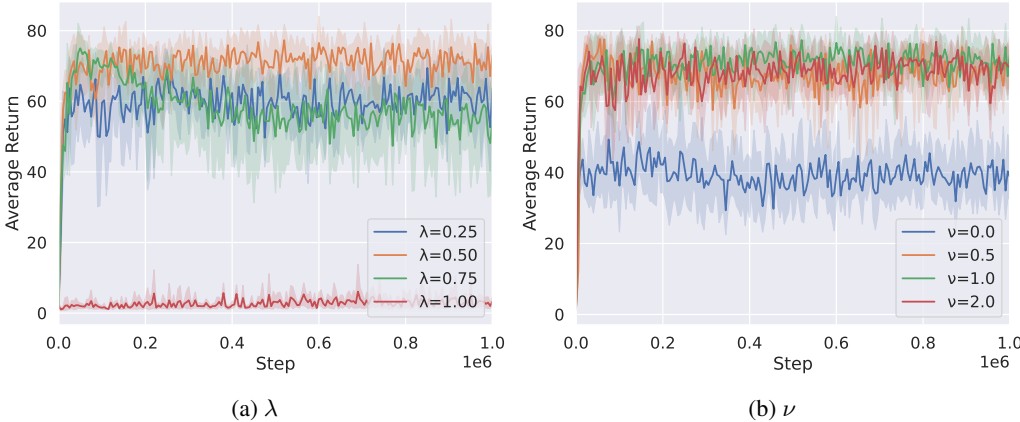

(a) $\lambda$        (b) $\nu$

Figure 6: Ablation study results on the hopper-medium-replay dataset. (a) and (b) each shows the average performance results for DPPO while varying $\lambda$ and $\nu$.

Table 2 shows the results on the more challenging D4RL Adroit pen and Kitchen tasks. Once again, DPPO outperforms all the baselines by a large margin. Especially, the performance gap is the largest in the Adroit pen tasks, where DPPO even surpasses the methods that learn with ground-truth rewards. A major distinguishing factor between the Adroit pen and the other environments is the dimensionality of the action space, which amounts to 24, significantly larger than the action spaces ranging from 3 to 9 in other environments. We conjecture that the value learning approach of the baseline methods struggles to scale up to the high-dimensional action spaces of Adroit. Furthermore, DPPO outperforms the baselines on the Kitchen tasks as well, underscoring the scalability of our method to tackle more complex tasks.

### 4.3 Ablation studies

We assess the importance of the two crucial components of DPPO, which are the conservativeness regularizer $\lambda$ and the smoothness regularizer $\nu$. The ablation results are presented in Figure 6. The results show that both components are crucial for achieving high performance. Moreover, the experiments demonstrate that DPPO exhibits a considerable degree of robustness to variations in the strength of the regularizations. Since our smoothness regularizer can also be easily applied to reward-modeling baselines, we evaluate how applying this regularizer to the baselines affects the overall performance in Appendix D. While the smoothness regularizer does improve the performance of the baselines, the improved performance falls behind DPPO.

### 4.4 Effect of dataset size

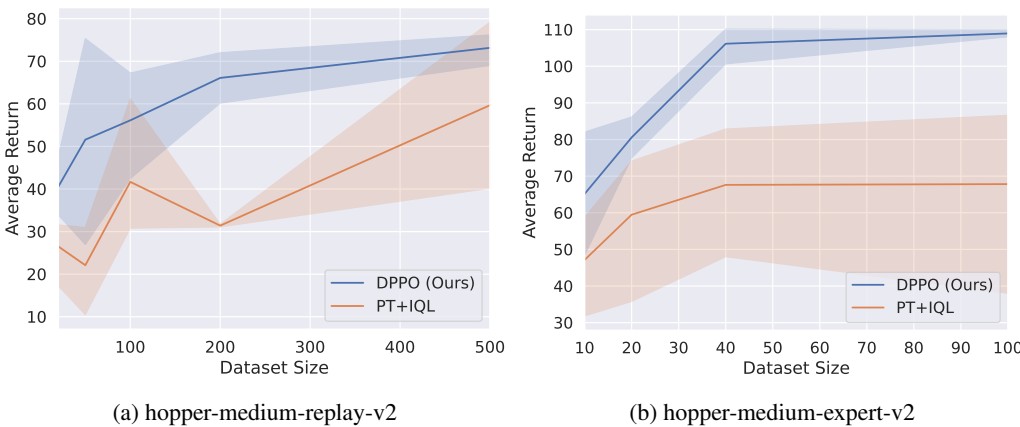

(a) hopper-medium-replay-v2        (b) hopper-medium-expert-v2

Figure 7: Average return results of each method while varying the size of the preference dataset.

Previous works primarily focus on evaluating their algorithms using a fixed preference dataset for each task [27, 13]. In this section, we evaluate how the size of the preference dataset impacts the overall performance of PbRL algorithms. Concretely, we vary the size of the preference dataset in the hopper-medium-replay and hopper-medium-expert tasks and measure the performance of PbRL methods on each dataset size. The results are shown in Figure 7. We observe that DPPO consistently outperforms the baseline method on all dataset sizes, displaying higher average performance. Interestingly, on hopper-medium-replay, we find that the baseline method PT+IQL falls into a unique failure mode when the dataset size is 200, which is to stand still and not move forward.

## 4.5 Experiments with scripted teachers

Following prior works [10, 31, 32], we additionally evaluate our algorithm using preference labels generated by scripted teachers. A scripted teacher is a synthetic teacher who adheres strictly to the task rewards when making decisions: $\sigma^0 \succ \sigma^1 \Leftrightarrow \sum_{t=0}^{k} r_t^0 > \sum_{t=0}^{k} r_t^1$. It is important to note that the scripted teacher setting is not as realistic as the human teacher setting, and we include this section for reference purposes. The results in Figure 8 show that our algorithm continues to outperform the baselines in terms of aver-

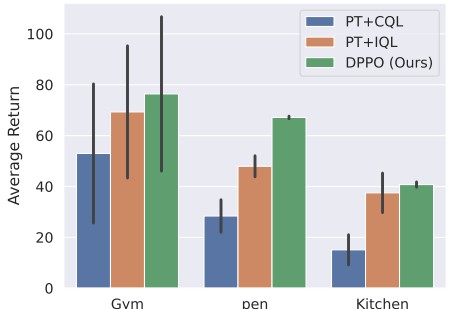

Figure 8: Average performance on the scripted teacher setting.

age performance, even in this synthetic setting. In detail, the results for DPPO and PT+IQL resemble the results from the human teacher setting. However, the performance of PT+CQL has dropped significantly compared to the human teacher setting. This disparity of performance reassures the observations from [27] that preferences from human teachers and scripted teachers have different characteristics and should not be treated interchangeably.

## 4.6 Fine-tuning LLMs with DPPO

Recent works show that PbRL can be used to fine-tune large language models with human preference, which is commonly termed RLHF (reinforcement learning from human feedback) [52, 39, 4]. In RLHF, a reward model is trained to predict human preference between two output texts, and then the learned reward model is employed to fine-tune the language model through off-the-shelf RL algorithms such as PPO. As from offline RL, we can fine-tune a language model directly with the preference predictions by replacing PPO with DPPO. This replacement allows removing unnecessary assumptions required to run reward-based policy optimization techniques like PPO, for example assuming the reward is given only at the end of the output sequence. To empirically examine if DPPO can be used to fine-tune large language models, we conducted some preliminary experiments using human preference datasets to train the preference models and human evaluation to evaluate the tuned language model. We refer to Appendix E for more details regarding the implementation details and the experimental setup.

Table 3: RLHF results using DPPO (ours) compared to PPO. The values in the parentheses denote the gain on average reward compared to the original model.

| Fine-tuning method | Avg. reward (↑) | KL divergence (↓) | Human eval. win rate (↑) |
|---|---|---|---|
| PPO | 4.335 (+1.192) | 0.0091 | 0.667 |
| DPPO (Ours) | **4.515 (+1.372)** | **0.0083** | **0.697** |

The results in Table 3 show that DPPO can be successfully used to fine-tune large language models with real human preference, achieving performance comparable to the reward-based RLHF method. We believe that this result is promising as it shows that DPPO can be used to fine-tune large language models without the need for reward modeling.

# 5 Related works

**Preference-based reinforcement learning**    A long line of works has studied learning agents from human preference [1, 41, 12, 3, 47]. Notably, [10] showed that we can scale PbRL by learning a reward model from preference and applying it to off-the-shelf RL algorithms. Several follow-up works have been proposed building upon this work. For example, a work merges the PbRL framework with imitation learning by introducing other types of supervision such as demonstrations [21]. Other works utilize techniques from semi-supervised learning or data augmentation to improve sample efficiency [31, 40]. Another line of work aims to improve the reward model by removing Markovian assumptions [13, 27]. Recent works have shown that PbRL can greatly boost the performance of large-scale language models by finetuning them with human feedback [52, 56, 37, 39, 45, 38]. Similar to our work, [55] leverages the preference information to directly optimize the policy using the concept of trajectory distance. However, their method requires the two trajectories to start from the same initial state and to roll out from the current policy being trained, which makes it challenging to utilize extensive pre-collected data. Concurrent to our work, [25, 43] also explore the idea of directly optimizing the policy with preference information.

**Offline reinforcement learning**    Offline reinforcement learning methods adopt various techniques to bias the learned policy towards the given offline dataset. For example, some works directly regularize the policy to prefer the actions from the dataset [29, 57, 17]. Another line of works implicitly biases the policy by regularizing the value function [30, 2, 58]. Some works instead start from a SARSA-style TD-update algorithm to avoid querying values for out-of-distribution actions [7, 28]. A more recent line of work casts the offline RL problem as a sequence modeling problem, where a model is learned to generate actions conditional to the given target return [8, 22, 33, 46, 14]. This approach soon gained popularity in the literature due to its high stability and scalability compared to the more traditional value-based approaches [33, 46].

**Contrastive learning**    A core design choice for contrastive learning is to define the positive and negative sample pairs. An earlier work utilizes the structure of the data and considers two different patches from a single sample as a positive pair [53]. The most popular approach is to apply a heavy data augmentation to a single sample repeatedly to produce positive pairs [9, 20]. The success of these methods in unsupervised vision representation learning has inspired many follow-up works, such as application to supervised learning or extension to other domains such as NLP or graph [26, 42, 59].

In the RL domain, there also have been works that leverage contrastive learning to learn unsupervised representations tailored for RL [51]. Also, a recent work directly casts the contrastive learning framework as a goal-conditioned RL problem [15].

# 6 Discussion

Our proposed PbRL algorithm enables agents to learn directly from preference signals, removing the need for reward modeling. We achieve this by formulating a new policy optimization problem under the contrastive learning framework. Thorough empirical evaluations on various offline PbRL tasks with actual human rewards show our method outperforms the baselines in most of the tasks considered. Interestingly, our algorithm shows better scalability to high-dimensional control tasks when compared to other RL baselines, including those that learn with ground-truth reward information. Additionally, our algorithm demonstrates better data efficiency. These results show that directly utilizing preference signals without reward modeling is a promising direction for PbRL.

A limitation of our work is that label noise, which is likely to exist for human evaluations, is not modeled in our algorithm. Investigating the effect of label noise stemming from human teachers would be an interesting direction for future research [23]. Also, our current algorithm does not incorporate the prediction confidence information, which was excluded as the empirical performance is already strong compared to the baselines. How to appropriately incorporate the prediction confidence would also be a meaningful research direction.

## Acknowledgements

We would like to express our gratitude to Deokjae Lee for his assistance in our natural language domain experiments. This work was supported by SNU-NAVER Hyperscale AI Center, Institute of Information & Communications Technology Planning & Evaluation (IITP) grant funded by the Korea government (MSIT) [No. 2020-0-00882, (SW STAR LAB) Development of deployable learning intelligence via self-sustainable and trustworthy machine learning, 55% and No. 2022-0-00480, Development of Training and Inference Methods for Goal-Oriented Artificial Intelligence Agents, 25%], and Basic Science Research Program through the National Research Foundation of Korea (NRF) funded by the Ministry of Education (RS-2023-00274280), 20%. Hyun Oh Song is the corresponding author.

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

# A   Human preference datasets

For the D4RL Gym hopper, walker2d, and Adroit pen tasks, we use the human preference datasets provided by [27]. Each dataset consists of 100 preference triples, except for the Gym *-medium-replay datasets which contain 500 preference triples[3]. Each trajectory segment is of length 100 (*i.e.*, k=100). For the other tasks such as Gym halfcheetah and Kitchen, we could not find publicly available human preference datasets. Therefore, we generate new preference datasets with human evaluation by strictly following the procedure from [27]. The preference labeling process consists of two steps: sampling a pair of trajectory segments from the dataset and labeling their preference under some predefined guidelines. For the first step, [27] does not specify how the segments were sampled, so we first randomly sample two trajectories from the offline dataset and then randomly choose a segment from each of the two trajectories to form a segment pair. We provide more details of the labeling process for each task below.

## A.1   Gym HalfCheetah

Following the experiment settings from [27], we sample 500 and 100 segment pairs from halfcheetah-medium-replay and halfcheetah-medium-expert datasets, respectively. We assign the human teachers (the authors) to make preference decisions under the following guidelines:

- The primary goal for the agent is to move forward as far as possible.

- If the two agents both satisfy the primary goal similarly, the agent that has a more stable posture is preferred. In other words, prefer the agent that staggers less.

## A.2   Franka Kitchen

Franka Kitchen environment provides six available sub-tasks, where the goal is to perform four of them in a designated order: open the microwave oven, move the kettle to the top left burner, turn on the light switch, and open the slide cabinet. The other two tasks, which are turning on the burner switch and opening the hinge cabinet, are dummy tasks without any positive reward. Figure 9 shows an example of a goal-related task and a dummy task. The robot on the left is moving a kettle, which is a goal-related task. On the other hand, the robot on the right is turning on a burner, which is a dummy task.

For each offline dataset (kitchen-partial and kitchen-mixed), we sample 100 pairs of trajectory segments and label them under the following guidelines:

- An agent that solves more goal-related tasks (in order) is preferred.

- If the two agents solved the same number of goal-related tasks, the agent that solved fewer dummy tasks is preferred.

- If the two agents solved the same number of goal-related and dummy tasks, the agent that was working on another goal-related task is preferred.

---

[3]For walker-medium-replay, [27] originally uses a preference dataset of size 500, but we subsample 100 samples from this dataset as we find that a smaller dataset suffices for PbRL algorithms to achieve reasonable performance.

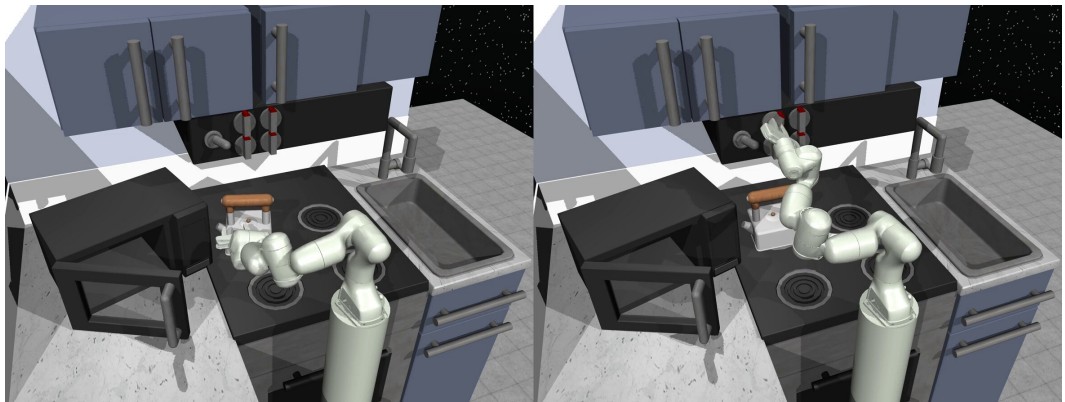

Figure 9: Example of a goal-related task (left) and a dummy task (right) in Franka Kitchen. The task on the left is to move a kettle and the task on the right is to turn on a burner.

# B    Offline RL experiment details

For each method, we evaluate its performance by collecting ten trajectories with online interaction and measuring the average return. Following the standard protocol of the offline RL literature, we rescale the average return by $100 \cdot \frac{R - R_{\text{random}}}{R_{\text{expert}} - R_{\text{random}}}$, where $R$ denotes the (empirical) average return of the policy, and $R_{\text{random}}$ and $R_{\text{expert}}$ are the expected returns of a random policy and an online expert policy, respectively. We provide the implementation details for each PbRL method below.

## B.1    PreferenceTransformer (PT)

We use the official implementation from the authors to train the reward model[4]. Then, for PT+IQL, we run IQL with the default hyperparameters for each environment using the official code[5]. Similarly, for PT+CQL, we run CQL with the default hyperparameters for each environment using the official code[6]. Table 4 and Table 5 lists the detailed hyperparameter settings for IQL and CQL.

Table 4: Hyperparameter settings for IQL in PT+IQL.

| Task name | Learning rate | # Layers | Hidden layer size | $\tau$ | $\beta$ | Dropout |
|---|---|---|---|---|---|---|
| Gym | $3 \cdot 10^{-4}$ | 2 | 256 | 0.7 | 3.0 | - |
| Adroit pen, Kitchen | $3 \cdot 10^{-4}$ | 2 | 256 | 0.7 | 0.5 | 0.1 |

Table 5: Hyperparameter settings for CQL in PT+CQL.

| Task name | Learning rate | # Layers | Hidden layer size | $\alpha$ | $\tau$ | Min Q Ver. |
|---|---|---|---|---|---|---|
| Gym | $1 \cdot 10^{-4}$ | 3 | 256 | 10.0 | - | 3 |
| Adroit pen | $3 \cdot 10^{-4}$ | 3 | 256 | 1.0 | 5.0 | 2 |
| Kitchen | $3 \cdot 10^{-4}$ | 3 | 256 | 1.0 | 5.0 | 3 |

## B.2    DPPO (Ours)

For the preference predictor, we start from a GPT-2 based transformer architecture from PT and apply some modifications. First, we removed the preference attention layer proposed by PT. Also, since PT requires per-step reward predictions, their transformer output is restricted to a scalar value for each

---

[4] https://github.com/csmile-1006/PreferenceTransformer
[5] https://github.com/ikostrikov/implicit_q_learning
[6] https://github.com/aviralkumar2907/CQL

timestep. Instead, our preference model outputs a vector embedding per step, which is aggregated and forwarded into a final MLP layer for the preference score prediction. We train the predictor for 10,000 update steps on the preference dataset. Table 6 lists more details of the training of the preference predictor. For policy optimization, we use a 2-layer MLP architecture and update the policy for 1e6 update steps, following the experimental protocol of IQL. Unlike CQL and IQL which adopt a stochastic policy model, we use a deterministic policy model as we find that both models have no difference in empirical performance. More detailed settings for the policy optimization procedure are specified in Table 7.

Table 6: Hyperparameter settings of the preference predictor training process in DPPO (Ours).

| Task name | Learning rate | # Layers | embedding dim | $\nu$ | $m$ |
|---|---|---|---|---|---|
| All tasks | $1 \cdot 10^{-4}$ | 1 | 256 | 1.0 | 20 |

Table 7: Hyperparameter settings of the policy optimization process in DPPO (Ours).

| Task name | Learning rate | # Layers | layer size | $\lambda$ | Dropout |
|---|---|---|---|---|---|
| Gym | $3 \cdot 10^{-4}$ | 2 | 256 | 0.5 | 0.25 |
| Adroit pen, Kitchen | $3 \cdot 10^{-4}$ | 2 | 256 | 0.1 | 0.25 |

## C  Offline RL results with more baselines

Our offline RL experiments in the main paper focus on baselines that apply value-based RL algorithms after modeling the reward, following [27]. However, another interesting line of work in offline RL is the behavior cloning-based RL algorithms, which typically learn a policy conditional to a target return or goal [8, 14]. Here, we provide the full experiment results including two behavior cloning-based baselines: %BC, which performs behavior cloning on trajectories with top-10% return values from the offline dataset, and RvS [14].

Table 8: Normalized average return on D4RL Adroit pen and Kitchen tasks, averaged over 5 seeds. $\pm$ denotes the standard deviation.

| Task Name | PT+%BC | PT+RvS | PT+CQL | PT+IQL | DPPO (Ours) |
|---|---|---|---|---|---|
| pen-human | $19.4 \pm 6.7$ | $-1.8 \pm 0.5$ | $31.6 \pm 3.3$ | $53.0 \pm 31.7$ | $\mathbf{76.3 \pm 14.4}$ |
| pen-cloned | $37.4 \pm 7.5$ | $-2.2 \pm 0.3$ | $18.3 \pm 10.6$ | $42.9 \pm 24.4$ | $\mathbf{75.1 \pm 7.7}$ |
| Average | 28.4 | -2.0 | 25.0 | 48.0 | **75.7** |
| kitchen-mixed | $40.9 \pm 9.0$ | $27.5 \pm 9.4$ | $12.3 \pm 7.7$ | $48.0 \pm 11.9$ | $\mathbf{52.5 \pm 3.1}$ |
| kitchen-partial | $53.4 \pm 9.0$ | $26.0 \pm 4.9$ | $14.1 \pm 13.0$ | $40.2 \pm 12.3$ | $\mathbf{49.4 \pm 5.7}$ |
| Average | 47.2 | 26.8 | 13.2 | 44.1 | **51.0** |

The results in Table 8 show that DPPO outperforms the behavior cloning-based baselines on all tasks, with the performance gap being especially large on the challenging pen tasks.

## D  Additional ablation on the smoothness regularizer

In recognition of the fact that our novel smoothness regularizer can also be integrated into any reward-modeling baselines, we assess how its application to the baselines would influence the overall performance and present results in Table 9.

Table 9: Normalized average return on D4RL Adroit pen and Kitchen tasks, averaged over 5 seeds. $\pm$ denotes the standard deviation.

| Task Name | PT+CQL | PT+CQL+$\nu$ | PT+IQL | PT+IQL+$\nu$ | DPPO (Ours) |
|---|---|---|---|---|---|
| pen-human | $31.6 \pm 3.3$ | $18.3 \pm 17.2$ | $53.0 \pm 31.7$ | $53.7 \pm 42.3$ | $\mathbf{76.3 \pm 14.4}$ |
| pen-cloned | $18.3 \pm 10.6$ | $32.7 \pm 11.2$ | $42.9 \pm 24.4$ | $49.8 \pm 32.2$ | $\mathbf{75.1 \pm 7.7}$ |
| Average | 25.0 | 25.5 | 48.0 | 51.8 | **75.7** |
| kitchen-mixed | $12.3 \pm 7.7$ | $12.0 \pm 5.0$ | $48.0 \pm 11.9$ | $49.4 \pm 5.2$ | $\mathbf{52.5 \pm 3.1}$ |
| kitchen-partial | $14.1 \pm 13.0$ | $11.4 \pm 11.2$ | $40.2 \pm 12.3$ | $\mathbf{49.4 \pm 5.2}$ | $\mathbf{49.4 \pm 5.7}$ |
| Average | 13.2 | 11.7 | 44.1 | 49.4 | **51.0** |

Although our smoothness regularizer does enhance the baseline methods' performance, this improved performance still lags behind our proposed method. This outcome suggests that the process of directly optimizing the policy through the use of preference information plays a critical role in the overall performance.

# E   RLHF experiment details

The fine-tuning procedure of RLHF is similar to the PbRL process discussed in our paper. First, a preference predictor is trained to assign high scores to texts that are more preferred by human teachers. Then, the reward for fine-tuning a language model is defined by combining the preference predictor and a regularizer on policy (language model) shift. Concretely, $r := r_\phi - \xi r_{\text{KL}}$, where $r_\phi$ is the output of the preference predictor and $r_{\text{KL}}$ is the KL divergence between the output distributions of the fine-tuned model and the original (pretrained) model. This reward function aims to guide the fine-tuned model to generate outputs that are more preferred by human teachers while not deviating far from the original model. After defining the reward function, PPO [48] is applied to optimize the language model to maximize the expected reward. Recent works show that this fine-tuning process can effectively align the language models with human preference, for example by generating more helpful and harmless sentences.

We can define a DPPO-style score metric that resembles the reward function from the original RLHF process as

$$S_{\text{RLHF}}(\theta; \mathcal{D}, \phi) := \mathop{\mathbb{E}}_{\sigma^0, \sigma^1 \sim \pi_\theta(\cdot | \mathbf{x}),\ \mathbf{x} \sim \mathcal{D}} \left[ (1 - \widehat{y}) \cdot s\left(\pi_\theta, \sigma^0, \sigma^1\right) + \widehat{y} \cdot s\left(\pi_\theta, \sigma^1, \sigma^0\right) \right]$$
$$- \xi \mathop{\mathbb{E}}_{\mathbf{x} \sim \mathcal{D}} \left[ D_{\text{KL}}(\pi_\theta(\cdot | \mathbf{x}), \pi_{\theta_0}(\cdot | \mathbf{x})) \right]$$
$$\text{s.t.} \quad \widehat{y} = \mathbb{1}\left\{ \widehat{P}\left[\sigma^0 \succ \sigma^1; \phi\right] > 0.5 \right\},$$

where $\mathcal{D}$ is a prompt dataset, $\sigma^i$ is the output sequence sampled from the fine-tuned language model $\pi_\theta$, $\pi_{\theta_0}$ is the original pretrained language model, and $D_{\text{KL}}$ is the KL divergence. Here we remove the regularizer $\lambda$ as the KL divergence term naturally prevents the policy from deviating far from the preferred sequences. Now we can directly fine-tune the policy by maximizing this score metric.

We evaluate our resulting RLHF algorithm by fine-tuning a pretrained OPT-1.3b language model which has 1.3 billion trainable parameters [60]. To train the preference predictor, we use the HH-RLHF dataset which contains 161K pairs of human preference data about helpfulness and harmlessness [4]. We choose OPT-350m with 350 million trainable parameters as the architecture of the predictor model. We evaluate our RLHF algorithm by comparing it with the conventional RLHF that uses the same preference predictor model and performs PPO. Our algorithm was implemented on DeepSpeed-Chat, an open code base for RLHF training[7].

We consider several evaluation metrics to measure the performance of our RLHF algorithm. First, we calculate the expected reward, which measures the empirical mean of $r_\phi$ from the outputs of the fine-tuned model. Second, we measure the KL divergence between the outputs of the fine-tuned model and the original model. These two metrics evaluate how well the two competing objectives

---

[7]`https://github.com/microsoft/DeepSpeedExamples/tree/master/applications/`
`DeepSpeed-Chat`

(maximizing the alignment with human preference and staying close to the original model) are optimized. Additionally, we perform an actual human evaluation on whether the fine-tuned models are more preferred than the original model. In detail, for a randomly sampled prompt from $\mathcal{D}$, we generate two responses each from the fine-tuned model and the original model. Then, we ask three human workers to choose which response they prefer more in terms of helpfulness, and designate the final preferred response by majority voting[8]. We repeat this trial 300 times for each fine-tuning method and measure the win rate of the fine-tuned models.

The evaluation results in Table 3 show that our RLHF algorithm achieves a higher average reward with lower KL divergence from the original model when compared to the conventional RLHF using PPO. This shows that DPPO exhibits a better trade-off between human preference alignment and closeness to the original model. Also, DPPO achieves a higher win rate versus the original model on actual human evaluation, which shows the fine-tuned model from our algorithm is more aligned with human preference. These results demonstrate that DPPO is a promising approach for addressing preference-based learning problems in NLP.

## F    Discussion on the Antmaze environment in D4RL

We acknowledge that many offline RL studies highlight experiment results from the D4RL Antmaze task to demonstrate the efficacy of their methods in more challenging environments. Yet, it must be noted that the official implementation of Antmaze has a critical bug in its goal-setting procedure[9]. Concretely, the offline datasets of Antmaze assume a multi-goal setting, where a random goal location is chosen at the start of each episode, while the actual environment (unintentionally) uses a single fixed goal for every episode. Consequently, the optimal policies with regard to the offline dataset and the environment differ: According to the offline dataset, the optimal policy is to quickly sweep through all the goal candidates, while within the environment context, the optimal policy is to move directly to the solitary fixed goal.

To fix this, we can modify the Antmaze environment to choose a random goal location for each episode, thereby aligning with the provided offline dataset. Surprisingly, after this modification, we observe that current offline RL algorithms struggle to achieve meaningful performance:

Table 10: Performance of offline RL algorithms on the original and fixed Antmaze environment.

| Task Name | Orig. Antmaze | | | Fixed Antmaze | | |
|---|---|---|---|---|---|---|
| | %BC | CQL | IQL | %BC | CQL | IQL |
| medium-play-v2 | $48.7 \pm 3.3$ | $44.7 \pm 27.8$ | $62.0 \pm 4.1$ | $10.5 \pm 2.9$ | $9.8 \pm 4.8$ | $1.5 \pm 1.5$ |
| medium-diverse-v2 | $35.5 \pm 6.0$ | $24.4 \pm 32.3$ | $66.2 \pm 6.3$ | $10.4 \pm 3.2$ | $3.1 \pm 3.3$ | $1.8 \pm 0.8$ |
| large-play-v2 | $12.2 \pm 7.5$ | $11.7 \pm 11.1$ | $59.8 \pm 8.1$ | $11.1 \pm 1.4$ | $5.5 \pm 3.1$ | $14.5 \pm 4.7$ |
| large-diverse-v2 | $10.8 \pm 3.6$ | $5.5 \pm 9.0$ | $55.2 \pm 6.5$ | $8.0 \pm 0.8$ | $4.8 \pm 1.3$ | $11.2 \pm 1.6$ |

Due to this stark difference in the evaluation performance, we decided not to include the Antmaze results in our main paper. We hope the offline RL community recognizes this problem in future studies involving Antmaze.

## G    Resources

All our offline RL experiments were run on a single RTX 3090 GPU with 10 CPU cores (Intel(R) Xeon(R) Gold 5218R CPU @ 2.10GHz). For the RLHF experiments, we used an A100 GPU with 10 CPU cores (AMD EPYC 7402 24-Core Processor).

---

[8]We used Amazon MTurk `https://www.mturk.com/` to conduct our human evaluation experiments.
[9]`https://github.com/Farama-Foundation/D4RL/issues/142`

## H  Broad impact

Our proposed method can be used to train RL agents that better align with human preferences. This method has the potential to significantly facilitate the deployment of AI services that cater to the specific requirements of the general public. However, it is essential to recognize that the learned RL agent can be biased toward certain values (*e.g.*, races, ethnic groups, religions, ...) if the preference dataset itself exhibits biases toward those values. Therefore, practitioners should exercise caution and thoroughly examine their preference dataset to identify and address any potential biases.

