# OpenReview forum: "Direct Preference-based Policy Optimization without Reward Modeling"
_NeurIPS.cc/2023/Conference — NeurIPS 2023 poster_

### Official Review · Reviewer_E68m · 2023-06-26

**Soundness:** 2 fair
**Presentation:** 3 good
**Contribution:** 2 fair
**Rating:** 7
**Confidence:** 5

**Summary:**

This paper investigates learning a policy directly from preferences, i.e., pairwise comparison between trajectory segments. By defining a metric between policy and trajectory segments, the authors show that this problem resembles contrastive learning problems, and they propose to learn a policy by minimizing the distance with preferred trajectory segments. The author verified the proposed approach using D4RL benchmarks.

**Strengths:**

1. The storytelling is concise and relevant. Materials are organized in a way that facilitates reading.
2. The proposed policy scoring metric is intuitive and novel.

**Weaknesses:**

1. Discussion and comparison with some very important alternative methods are missing, which weakens the significance of this paper.
For example, the problem of learning policies from preferences directly was studied by Liu et al. [1] before. Although they carried out their experiments in an online PbRL setting, it is straightforward to use their approach in the setting of this paper. Moreover, the proposed approach utilizes both offline data and preferences in learning. Two studies in the literature [2,3] also work in the same setting. Although the authors mentioned [2] in the related work section, they did not include results for the approach proposed in [2] or [3]. Importantly, the proposed approach outperforms the preference transformer mainly on medium-expert datasets, which are of high quality. Missing comparison with the imitation-based approach [2,3] undermines the significance of experiments.

2. There lack of discussion on the connection between the proposed approach and behavior cloning. In fact, it was not straightforward for me to understand why minimizing a distance metric leads to a policy when reading sec 3.2 for the first time. It became clear after I realized the connection between $$d(\pi,\sigma^i)$$ and the objective of behavior cloning.


[1] Meta-Reward-Net: Implicitly Differentiable Reward Learning for Preference-based Reinforcement Learning, NeurIPS 2022.

[2] Reward learning from human preferences and demonstrations in Atari, NeurIPS 2018.

[3] Conﬁdence-Aware Imitation Learning from Demonstrations with Varying Optimality, NeurIPS 2021.

**Questions:**

1. I would suggest the authors compare the proposed approach with [1] to show the efficacy of their method for bypassing reward learning.

2. The comparison with [2,3] is also relevant, especially for datasets with high quality.

3. While the connection between PbRL and contrastive learning is interesting, the objective (eq. (4)) is an extension of the Bradley—Terry model. In other words, the output policy is not pulled away from MULTIPLE negative samples. Since the authors made use of unlabeled trajectories in a preference labeler (eq (5)), I think it is insightful to discuss a more close utilization of contrastive learning for PbRL.

4. I would suggest the authors discuss the connection between their approach and behavior cloning.

**Limitations:**

The authors discussed some technical limitations of their approach, but there lacks a discussion for broader impact or social impact.

---

> ### Author Rebuttal · Authors · 2023-08-09
>
> We thank the reviewer for the encouraging comments and constructive feedback. We would like to address your comments below.
>
> **1. Discussion and comparison with some very important alternative methods are missing**
>
> Thank you for pointing out some relevant baselines for our work. While [1] does model the preference directly (using the Q-value as the substitute for the preference score), it maintains an assumption that an underlying per-step reward structure is present and leans on this structure to facilitate preference learning. We made some minor modifications to [1, 2] to adapt them to the offline context (e.g., changing the policy optimization algorithm from SAC to IQL on [1]), and present the results below:
>
> |  | [1] | [2] | DPPO (Ours) |
> |---|---|---|---|
> | pen-human | 57.5 $\pm$ 18.3 | 38.6 $\pm$ 12.8 | **76.3 $\pm$ 14.4** |
> | pen-cloned | 45.3 $\pm$ 26.5 | 30.8 $\pm$ 24.3 | **75.1 $\pm$ 7.7** |
> | Average | 51.4 | 34.7 | **75.7** |
> | kitchen-mixed | 38.8 $\pm$ 13.8 | 23.8 $\pm$ 2.5 | **52.5 $\pm$ 3.1** |
> | kitchen-cloned | 33.8 $\pm$ 7.5 | 38.8 $\pm$ 18.5 | **49.4 $\pm$ 5.7** |
> | Average | 36.3 | 31.3 | **51.0** |
>
> The results show that DPPO significantly outperforms the considered baselines.
>
> (Note: Regarding [3], we discovered that the official implementation's package configuration is not compatible with the Adroit and Kitchen environments. We plan to address this matter during the rolling discussion phase.)
>
> **2. lack of discussion on the connection between the proposed approach and behavior cloning**
>
> Thank you for the feedback. Broadly speaking, our proposed method can be thought of as imitating the behavior from more desirable trajectory segments while distancing itself from the behavior in less desirable segments. We will make this clearer in the updated manuscript. In addition, we have conducted an empirical comparison with some of the BC-based methods, such as RvS and filtered BC, in common response 4. The findings reveal that DPPO substantially excels over the BC-based PbRL approaches.
>
> **3. In other words, the output policy is not pulled away from MULTIPLE negative samples**
>
> Thank you for the insightful comment. We would like to point out that the decision to use a single negative sample was driven by computational efficiency. Our findings indicate that the number of negative samples doesn't significantly influence empirical performance (we plan to include the full comparison results as soon as possible). We will expand our discussion on the contrastive learning formulation in the updated manuscript. Meanwhile, could the reviewer be so kind as to elaborate further on what you were referring to with the statement `Since the authors made use of unlabeled trajectories using a preference labeler (eq (5))`?
>
> **4. lacks a discussion for broader impact or social impact.**
>
> Thank you for your observation. Due to the constraints on page length, we placed the broader impact section in the appendix (section E). We will incorporate this section into the main body of the updated manuscript.

---

> > ### Author Response · Authors · 2023-08-14
> > **Experiment Results on multiple negative samples**
> >
> > Hi,
> >
> > We finished running our multiple negative sample experiments (for question **3.** above). In detail, we modify our original policy score metric in  Eq (4) by the following procedure: (1) Given a batch of trajectory segments, a preference score is assigned to each segment using our preference model. (2) The segments are sorted in descending order of their preference scores. (3) Let's denote the sorted segments as $\\{\sigma^i\\}_{i=1}^n$. For $\sigma^i$, we set $\sigma^{i+1}, \ldots, \sigma^n$ to be the negative samples and apply contrastive learning loss, resulting in the following policy score metric:
> >
> > $$
> > \frac{1}{n-1}\sum_{i=1}^{n-1} \log \frac{\exp \left( -d\left(\pi, \sigma^i\right) \right)}{\exp \left( -d\left(\pi, \sigma^i\right) \right) + \sum_{j=i+1}^n \exp \left( -\lambda d\left(\pi, \sigma^j\right) \right)}
> > $$
> >
> > Using this new formulation which adopts multiple negative samples, we re-ran our pen and kitchen experiments and present the results below:
> >
> > |  | DPPO (orig) | DPPO (multiple negative samples) |
> > |---|---|---|
> > | pen-human | 76.3 $\pm$ 14.4 | 62.5 $\pm$ 10.8 |
> > | pen-cloned | 75.1 $\pm$ 7.7 | 83.0 $\pm$ 18.9 |
> > | Average | 75.7 | 72.8 |
> > | kitchen-mixed | 52.5 $\pm$ 3.1 | 52.0 $\pm$ 2.8 |
> > | kitchen-cloned | 49.4 $\pm$ 5.7 | 47.3 $\pm$ 7.7 |
> > | Average | 51.0 | 49.8 |
> >
> > We find that the performance level remains the same considering the standard deviations. Also, it is important to note that all the hyperparameters ($\lambda$, $\nu$, ...) were fixed to the original setting and not tuned at all. This result suggests that our objective effectively can be considered a contrastive learning objective.

---

> > > ### Comment · Reviewer_E68m · 2023-08-15
> > >
> > > Thanks for the responses. I have corrected the mentioned typos and adjusted my evaluation accordingly.

---

> > > > ### Author Response · Authors · 2023-08-21
> > > >
> > > > Thank you for the response!
> > > >
> > > > We are glad that our rebuttal has resolved your initial concerns. Thank you for the time and effort spent providing insightful feedback for our work.

---

### Official Review · Reviewer_MTAP · 2023-06-28

**Soundness:** 3 good
**Presentation:** 2 fair
**Contribution:** 2 fair
**Rating:** 5
**Confidence:** 4

**Summary:**

In the context of Preference-based RL (PbRL), the authors propose to learn a preference model that ranks trajectory segments without assuming that the model decomposes into a sum of rewards in an MDP sense. The motivation behind this choice is that the human teachers themselves might not follow a model of the sort, and hence modeling these preferences can be made simpler and more accurate without the usual reward decomposition assumption in PbRL.

Using this preference prediction model, the authors define a non-learned policy scoring function for ranking policies based on average similarity in action with trajectory segments in a dataset. The overall method is evaluated in an offline setting with a mix of labeled and unlabeled data. The preference model is trained on the labeled data, and used to predict preferences over the unlabeled dataset. Using these predictions, a policy optimizing the scoring function on the unlabeled dataset is returned. Comparisons with a recent baseline was provided on several robotics tasks.


**Strengths:**

- The hypothesis of many PbRL papers that humans provide preferences according to a Markovian reward might be too strong, and it's important to have papers exploring the validity of this hypothesis
- The method shows improvements over a recent baseline on several tasks by using a very simple policy scoring function


**Weaknesses:**

- Some references are missing, for example in line 80 the authors state that "While prior PbRL approaches adopt a two-step procedure involving the construction of a reward (...)", whereas there are in fact PbRL papers that follow a very similar idea of using trajectory distances to compare policies without a reward model, e.g. Wilson et al., A Bayesian Approach for Policy Learning from Trajectory Preference Queries, NeurIPS2012.

- While the main claim of the paper is to introduce a new model for predicting trajectory preferences that is not based on rewards, I was very surprised that a formal definition of this model is never provided in the paper! In the appendix, we learn that the implementation is based on PreferenceTransformer [25] that is in fact decomposing the score as a sum of (history dependent) rewards, so this is a bit confusing.

- The experiments are limited to the offline setting, which is fine, but then it becomes unfair to compare to methods that assume a reward decomposition structure of the preferences because while this assumption might be restrictive, in exchange RL can be used to collect new data by optimizing the current reward model. In contrast, it is unclear how well the Direct Preference-based Policy Optimization method would work in the absence of offline data, since simply maximizing the scoring function on the labeled data will probably amount to behavioral cloning and offer very little in terms of exploration.

**Questions:**

- If hat{P} is based on PreferenceTransformer, is it still learning a (history dependent) reward model? Why hat{P} was not formally defined in the paper?
- How would you perform policy optimization in the absence of offline data? How will this offline data be generated in a given RL problem?
- If your approach and the baseline use the same preference model, what contributes the most to the improved performance? The added regularizers or the optimization of the scoring function instead of IQL/CQL
- In the appendix you state that policy optimization is "following the experimental protocol of IQL". What does this mean?

**Limitations:**

A limitation section is included but the paper does not discuss the counter part of not having a reward model and not being able to use RL for policy optimization, which seems important to me.

---

> ### Author Rebuttal · Authors · 2023-08-09
>
> We express our gratitude to the reviewer for the inspiring remarks and valuable feedback. We would like to address your comments below.
>
> **1. Some references are missing.**
>
> Thank you for pointing out an important previous work related to our paper. For a detailed discussion on related works, please refer to common response 2.
>
> **2. Clarification of the preference model.**
>
> Thank you for the feedback. For an in-depth explanation of our preference model, please refer to common response 1.
>
> **3. it is unclear how well the Direct Preference-based Policy Optimization method would work in the absence of offline data.**
>
> Drawing from the work of [1], our algorithm can be adapted to the online context by incorporating entropy-maximization terms [2]. More specifically, we can modify our policy optimization objective in Equation (2) by introducing a regularizing term that maximizes the entropy of the output distribution of the current policy. We evaluated this modified version of DPPO on the offline training-online finetuning setting. To elaborate, we run DPPO using only the offline data for 1 million gradient updates and proceed with online fine-tuning through 0.2 million online interactions. The outcomes for both DPPO and PT+IQL are presented below:
>
> |  | IQL (offline) | IQL (0.2m) | DPPO (offline) | DPPO (0.2m) |
> |---|---|---|---|---|
> | pen-human | 53.0 $\pm$ 31.7 | 60.1 $\pm$ 15.6 | 76.3 $\pm$ 14.4 | **82.5 $\pm$ 5.3** |
> | pen-cloned | 42.9 $\pm$ 24.4 | 40.4 $\pm$ 30.2 | 75.1 $\pm$ 7.7 | **85.3 $\pm$ 8.4** |
> | Average | 48.0 | 50.3 | 75.7 | **83.9** |
> | kitchen-mixed | 48.0 $\pm$ 11.9 | 46.7 $\pm$ 7.8 | 52.5 $\pm$ 3.1 | **54.2 $\pm$ 1.4** |
> | kitchen-cloned | 40.2 $\pm$ 12.3 | **49.7 $\pm$ 5.4** | **49.4 $\pm$ 5.7** | **48.3 $\pm$ 6.2** |
> | Average | 44.1 | 48.2 | **51.0** | **51.3** |
>
> The findings indicate that DPPO is well-suited for deployment in online fine-tuning environments. Additionally, we'd like to highlight that experiments on LLM RLHF are included in Appendix Section C, representing another instance of an online fine-tuning context.
>
> **4. what contributes the most to the improved performance? The added regularizers or the optimization of the scoring function instead of IQL/CQL**
>
> Thank you for the feedback. As analyzed in common response 3, we found that our smoothness regularizer also positively influences all the considered methods. However, the highest performance is attained through direct policy optimization that incorporates preference information.
>
> **5. "following the experimental protocol of IQL". What does this mean?**
>
> To clarify, this statement note that we adopted the same neural network architecture and the optimizer from IQL. We will make this clearer in the updated manuscript.
>
> [1] Zheng et al., Online Decision Transformer, ICML 2022.
>
> [2] Haarnoja et al., Soft Actor-Critic: Off-Policy Maximum Entropy Deep Reinforcement Learning with a Stochastic Actor, ICML 2018.

---

> > ### Comment · Reviewer_MTAP · 2023-08-14
> >
> > Thank you for your response. I believe the additional experiment in response 3 answers well my question and I raise my score accordingly.
> >
> > It is interesting that RL/Q-values are not necessary for policy optimization and can be replaced with the optimization of a simple scoring function. I am still doubtful that this would work beyond the offline data setting, and the new experiment with online fine tuning does not alleviate these concerns. Can the method really learn a sequential decision making policy from human feedback from scratch? Even if on simple control tasks?

---

> > > ### Author Response · Authors · 2023-08-21
> > >
> > > Thank you for the response!
> > >
> > > We are glad that our initial rebuttal addressed your question well.
> > >
> > > Regarding whether DPPO would work in a purely online setting, we believe there is significant scope for future exploration in this direction. We regret that due to the limited timeframe of the discussion phase, we were not able to carry out a thorough empirical assessment in a purely online setting. However, it is important to highlight that the supervised learning-based approach (where DPPO lies) has its own advantages compared to the value learning-based approach (such as IQL). Specifically, the supervised learning-based approach removes the need for intricate implementation tricks required in traditional RL, such as double-Q learning or target networks, as it is free from the deadly-triad issue [1]. Moreover, empirically, the supervised learning-based approach tends to exhibit much more stability in performance compared to the value learning-based approach. Our results in Table 1 and 2 also show that DPPO shows a much lower variance in performance compared to the baselines. We would appreciate it if the reviewer could consider these advantages of our supervised learning-based approach.
> > >
> > > We would like to once again express our gratitude to the reviewer for the insightful feedback during the rebuttal period.
> > >
> > > [1] van Hasselt et al., Deep Reinforcement Learning and the Deadly Triad, arXiv:1812.02648, 2018.

---

### Official Review · Reviewer_93eM · 2023-07-02

**Soundness:** 3 good
**Presentation:** 3 good
**Contribution:** 3 good
**Rating:** 5
**Confidence:** 5

**Summary:**

This paper presents DPPO, a method for offline Preference-based Reinforcement Learning (PbRL). Rather than using an Offline RL algorithm, DPPO uses a supervised learning objective inspired from contrastive learning. A distance metric is defined, and the policy is trained to have lower distance with preferred segments than with un-preferred segments. To attain good performance, the authors learn a preference predictor (analogous to a reward model, but with the whole segment as input) with a smoothness regularizer and introduce a scale penalty to policy-segment distances, both of which correlate with performance. Their approach is evaluated on D4RL with preference labels collected from real humans.


**Strengths:**

The paper was very easy to read (I appreciate that!) Explanations of major topics were largely complete.

The idea of reformulating the PbRL problem using ideas from contrastive learning is interesting, and appears to perform well in practice for the chosen benchmarks.

That brings me to experimentation. I believe the authors did a good job providing results on a number of applicable benchmarks, and I appreciate the use of real human feedback. That being said, there are a few points I wish they addressed (see weaknesses), mainly related to increased understanding of why the proposed contrastive objective works well.

The authors also ablated both hyper-parameters they introduced.

I believe that this has the potential to be a very strong paper. However, there are a few oversights that I would like to see addressed listed in the weaknesses section. If these concerns are addressed and questions answered, I would be happy to revise my score.


**Weaknesses:**

**Is Contrastive Learning Important?**
I wish the authors included an ablation showing the importance of the contrastive learning part of DPPO versus the segment smoothing objective. The smoothness regularization introduced in Eq. 5 could be used with *any* preference based RL algorithm, and does not appear to be unique to “reward” modeling vs. “preference” modeling. It would be great if the authors could include an ablation that trains PT-IQL with the preference smoothness regularizer. That would show that the performance of DPPO is due to the unique choice of contrastive objective, instead of the smoothness regularizer which could be applied to any reward-modeling algorithm.

**Limitations of DPPO**
I found the discussion of limitations and experiments surrounding limitations to be lacking. The authors opt to use a contrastive objective instead of approximate dynamic programming, and did not evaluate on any datasets where stitching is necessary. Supervised RL objectives (AWR, contrastive RL, GCSL etc.) do well when great policies can be learned with a single step of policy improvement, or when trajectory stitching isn’t necessary. On most of the chosen benchmarks, particularly Androit and Franka Kitchen, BC can do surprisingly well! For example, a score of 54 can be achieved on Androit pen cloned with just BC. This trend holds for many datasets (See Table 2 of https://arxiv.org/pdf/2004.07219.pdf). At the same time, the authors do not include results on AntMaze, which is known to require trajectory stitching, and was also included in the PreferenceTransformer paper to which the authors directly compare.

I want to emphasize that it is OK if DPPO does not do as well as IQL-based PbRL algorithms on AntMaze. Including these results and an enlightening discussion on when DPPO will perform better or worse would incline me to consider raising my score, not lower it.

**Modeling Rewards vs. Preferences**

Personally, I think the framing of the paper to be potentially mis-leading, specifically the part about it being “without reward modeling”. Yes, the authors do not have a “reward” model, but they still use a preference predictor with the same architecture as Preference Transformer (Appendix B). In fact the distinction between modeling a non-Markovian Reward (CITATION) and “modeling preferences” seems small: both have a neural network that takes in a state action segment and produces scores – the difference being that one is at each step (reward) and one is at the end (preference). It would be great if the authors could place more emphasis on why this difference matters.

**Related Work**
I found the related work section to be lacking. In particular, the authors omit several foundational works in preference-based RL [1,2,3,4,5], and more recent works as well [6,7,8]. I’ve included an incomplete number of examples.

Second, I will not penalize the authors for this since these works were made public after submission, but two highly related works have recently been released. [9], published at ICML also takes inspiration from contrastive learning and applies it to offline PbRL. [10], a preprint, removes the reward network from preference-based RL algorithms.

[1] A. Wilson, A. Fern, and P. Tadepalli. A bayesian approach for policy learning from trajectory preference queries. In F. Pereira, C. J. C. Burges, L. Bottou, and K. Q. Weinberger, editors, Advances in Neural Information Processing Systems, volume 25. Curran Associates, Inc., 2012. URL https://proceedings.neurips.cc/paper/2012/file/ 16c222aa19898e5058938167c8ab6c57-Paper.pdf

[2]  D. Sadigh, A. D. Dragan, S. S. Sastry, and S. A. Seshia. Active preference-based learning of reward functions. In Proceedings of Robotics: Science and Systems (RSS), July 2017. doi: 10.15607/RSS.2017.XIII.053.

[3]  J. R. Lepird, M. P. Owen, and M. J. Kochenderfer. Bayesian preference elicitation for multiobjective engineering design optimization. Journal of Aerospace Information Systems, 12(10): 634–645, 2015.

[4] E. Biyik, N. Huynh, M. J. Kochenderfer, and D. Sadigh. Active preference-based gaussian process regression for reward learning. In Proceedings of Robotics: Science and Systems (RSS), July 2020.

[5] W Bradley Knox and Peter Stone. Tamer: Training an agent manually via evaluative reinforcement. In 2008 7th IEEE international conference on development and learning, pages 292–297. IEEE, 2008.

[6] Joey Hejna and Dorsa Sadigh. Few-shot preference learning for human-in-the-loop RL. In Conference on Robot Learning, 2022

 [7] Liu, Yi, et al. "Efficient preference-based reinforcement learning using learned dynamics models." arXiv preprint arXiv:2301.04741 (2023).

[8] Daniel Shin and Daniel S Brown. Offline preference-based apprenticeship learning. arXiv preprint arXiv:2107.09251, 2021

[9] Kang, Yachen, et al. "Beyond reward: Offline preference-guided policy optimization." arXiv preprint arXiv:2305.16217 (2023).

[10] Hejna, Joey, and Dorsa Sadigh. "Inverse Preference Learning: Preference-based RL without a Reward Function." arXiv preprint arXiv:2305.15363 (2023).


**Questions:**

Major:
1. **There appear to be inconsistent results in comparison to Preference Transformer**, the primary paper with which the authors compare. For example, Preference Transformer achieves a score of 84.5 on Hopper Medium Replay in their publication (higher than DPPO), but in this paper has a score of 59.7. The results are also off by 30 points for Walker Medium Replay! Why is this? This feels important to answer and address in the paper, especially since the authors in the appendix state to use the same codebase as PT.
2. How would DPPO perform on the antmaze benchmark? (See weaknesses)
3. How would other preference-based RL algorithms perform when using the preference smoothness regularizer (see Weaknesses)?

Minor:
1. Figure 2: “Reward model from PbRL” Which model is this? Is it the preference transformer or a Markovian model?
2. Is it true that $d_{sa}$ and AGG must be differentiable? I believe it to be the case given the method. This should be explicitly stated!


**Limitations:**

The authors have very little discussion on the limitations of their approach. I would have liked to see an explicit section on limitations with some of the experiments in the Weaknesses section included.

---

> ### Author Rebuttal · Authors · 2023-08-09
>
> We extend our appreciation to the reviewer for the positive remarks and insightful feedback. We would like to address your comments below.
>
> **1. Is contrastive learning important?**
>
> In common response 3, we incorporated the baseline algorithms' evaluation results when combined with our smoothness regularizer. Although our smoothness regularizer elevates the baselines' performance, DPPO still outperforms them by a large margin. This observation underscores that the process of direct policy optimization by leveraging preference information is pivotal to achieving high performance.
>
> **2. Experiments on Antmaze**
>
> We acknowledge that many offline RL studies highlight Antmaze results to demonstrate the efficacy of their methods in more challenging environments. Yet, it must be noted that the official implementation of Antmaze suffers from a serious flaw in the procedure of goal setting (https://github.com/Farama-Foundation/D4RL/issues/142). Concretely, the critical bug in Antmaze is that the offline dataset assumes a multi-goal setting, where a random goal location is chosen at the start of each episode, while the actual environment (unintentionally) uses a single fixed goal for every episode. Consequently, the optimal policies with regard to the offline dataset and the environment differ: According to the offline dataset, the optimal policy is to fastly sweep through all the goal candidates, while within the
>  environment context, the optimal policy is to move directly to the solitary fixed goal.
>
> To fix this, we can modify the Antmaze environment to choose a random goal location for each episode, thereby aligning with the provided offline dataset. Surprisingly, after this modification, we observe that offline RL algorithms struggle to achieve meaningful performance, even when provided with true reward information:
>
> |  | filtered BC | CQL | IQL |
> |---|---|---|---|
> | antmaze-medium-play-v2 | 10.5 $\pm$ 2.9 | 9.8 $\pm$ 4.8 | 1.5 $\pm$ 1.5 |
> | antmaze-medium-diverse-v2 | 10.4 $\pm$ 3.2 | 3.1 $\pm$ 3.3 | 1.8 $\pm$ 0.8 |
> | antmaze-large-play-v2 | 11.1 $\pm$ 1.4 | 5.5 $\pm$ 3.1 | 14.5 $\pm$ 4.7 |
> | antmaze-large-diverse-v2 | 8.0 $\pm$ 0.8  | 4.8 $\pm$ 1.3 | 11.2 $\pm$ 1.6 |
>
> This is the reason why we did not include the Antmaze results in our paper. We perceive this flaw in the benchmark as a substantial concern for the offline RL community, especially considering that many existing offline RL studies continue to utilize this erroneous version of Antmaze. We plan to incorporate this discussion in our revised manuscript.
>
> That said, we provide the offline PbRL results on the fixed version of Antmaze below:
>
> |  | PT+CQL | PT+IQL | DPPO (Ours) |
> |---|---|---|---|
> | antmaze-medium-play-v2 | 3.3 $\pm$ 4.7 | 3.2 $\pm$ 4.4 | 1.8 $\pm$ 1.8 |
> | antmaze-medium-diverse-v2 | 0.0 $\pm$ 0.0 | 1.5 $\pm$ 1.5 | 0.0 $\pm$ 0.0 |
> | antmaze-large-play-v2 | 0.0 $\pm$ 0.0 | 13.5 $\pm$ 5.4 | 10.0 $\pm$ 9.4 |
> | antmaze-large-diverse-v2 | 0.0 $\pm$ 0.0 | 6.7 $\pm$ 5.8 | 7.5 $\pm$ 9.6 |
>
> **3. Modeling Rewards vs. Preferences.**
>
> Thank you for the feedback. Please refer to the common response 1.
>
> **4. Related work.**
>
> Thank you for the feedback. Please refer to the common response 2.
>
> **5. Inconsistent results on PT.**
>
> We did use the authors' official implementation (https://github.com/csmile-1006/PreferenceTransformer) to obtain the PT results for our paper. However, as highlighted in this issue (https://github.com/csmile-1006/PreferenceTransformer/issues/3), reproducing the performance of PT is challenging. Furthermore, as per the author's remark, some other less significant problems exist, such as sensitivity to the selection of random seeds. We conjecture that these factors contribute to the inconsistent results on PT when compared to the original paper. We will include this information and additionally provide the original paper's results in the updated manuscript.
>
> **6. Which model was used for Figure 2?**
>
> As noted in the figure's caption, we used PT to produce the results.
>
> **7. Is it true that $d_{\text{sa}}$ and $\text{AGG}$ must be differentiable?**
>
> As we currently use a gradient descent-style optimizer (Adam) to optimize the policy model, $d_{\text{sa}}$ and $\text{AGG}$ should be differentiable. However, should we adopt $d_{\text{sa}}$ and $\text{AGG}$ such that differentiation is impossible, we could utilize other gradient-free optimization methods (e.g., evolutionary algorithms, gradient approximation, ...). We believe that this extension would be a valuable direction for future research! We will include this discussion in the updated manuscript.

---

> > ### Comment · Reviewer_93eM · 2023-08-10
> > **Thank you for your detailed response!**
> >
> > Thank you for your detail response to my concerns. I believe my primary concerns have been addressed and I am consequently raising my score.
> >
> > There is one additional point I would like to address -- after looking at the authors code it appears there is a discrepancy between how the authors present their method in the paper and the actual implementation:
> >
> > The authors present their method as using the binary classification loss.
> >
> > In practice, however, it seems like the authors are using the InfoNCE loss across all segments in the batch. It looks like the preference predict is used to assign continuous scores, segments are all sorted within a batch, and then the contrastive loss is computed based on the score with a weighting function, or something of that type. This trick seems like it would greatly increase the learning signal, and I assume contributes to performance. I would request that the authors include more details in both their rebuttal and paper on how the loss is actually computed.
> >
> > Thank you!

---

> > > ### Author Response · Authors · 2023-08-11
> > > **Thank you for the thoughtful response!**
> > >
> > > Thank you for your prompt response to our rebuttal and for raising the score!
> > >
> > > We are glad that our initial response has addressed your primary concerns. Also, we sincerely appreciate your time and effort spent for thoroughly evaluating our work, including the source code we provided.
> > >
> > > Before we delve into your additional concern regarding the actual implementation, for clarity, we would like to point out that our preference model does indeed employ the binary classification loss (please refer to `JaxPref/PrefTransformer.py` L617). The implementation the reviewer refers to seems to be the policy model's loss function (`learner.py` L23-67).
> > >
> > > As for the actual loss calculation of the policy model, it is correct that we assign a continuous preference score to each segment and compute the contrastive learning loss across every possible segment pair. Concretely, given a batch of segments $\\{\sigma^i\\}_{i=1}^n$, let's assume the segments are sorted in the descending order of their preference scores (obtained from our preference model). Our policy training loss for this batch is implemented as below:
> > >
> > > $$\sum_{1\leq i < j \leq n} s(\pi, \sigma^i, \sigma^j)$$
> > >
> > > To summarize, the contrastive learning loss term is calculated for every segment pair within the batch. Although this implementation might appear, at first look, to diverge from the original objective outlined in our paper (Eqn (2)), it's essential to note that this choice does not alter the update direction when the expectation is considered. Our intention with this implementation was to maximize the number of contrastive learning terms we could get from a given batch size.
> > >
> > > Regarding the weighting function (the reviewer might be referring to the `gap_weight_temperature` variable in `learner.py` L52), we apologize to the reviewer for the confusion. This variable stems from earlier experimental attempts that didn't make it to our method's final version. Currently, the variable is redundant and remains ineffective (with the weight consistently set at 1).
> > >
> > > In our revised version, we'll eliminate these redundant parts of the code and also elaborate further on the aforementioned implementation decision.
> > >
> > > Once again, we express our gratitude to the reviewer for the insightful feedback. Please let us know if you have any remaining questions!

---

### Official Review · Reviewer_CbsJ · 2023-07-07

**Soundness:** 2 fair
**Presentation:** 4 excellent
**Contribution:** 3 good
**Rating:** 7
**Confidence:** 3

**Summary:**

This paper presents an algorithm called DPPO that can learn a policy from paired trajectory comparisons without having to learn a reward function or do online rollouts. PBRL algorithms typically learn a reward function with a Boltzmann comparison model and then optimize it with an online RL algorithm, like PPO. In contrast, DPPO uses a kind of pseudo-Boltzmann model expressed in terms of difference in similarity to the learned policy rather than difference in returns. In particular, if trajectory $a$ is preferred to trajectory $b$, then the learned policy $\pi$ should be more similar to trajectory $a$, on average, than to trajectory $b$. DPPO is augmented with a BC-like regularizer that biases $\pi$ towards having low average divergence with the trajectories (which prevents it from having consistently high divergence over all trajectories), as well as an auxiliary loss term for learning from unlabeled trajectories (essentially the same as the main loss but with a 0.5 label for unlabeled trajectories).

Experiments on D4RL Gym, Adroit, and Franka Kitchen show DPPO typically obtains higher mean return than both an existing offline PBRL algorithm and two existing offline RL algorithms, even though the latter two have access to the ground truth reward function. This seems to be the case for both human and synthetic labels.

**Strengths:**

 - **[S1]** The need for online rollouts is a major practical obstacle to using reward-learning algorithms (e.g. IRL, PBRL) for control. Stable offline preference-based reinforcement learning algorithms would make it much easier to apply PBRL to systems where rollouts are expensive, since it is more difficult to tune such systems (examples include LLMs and real robots—although neither are used in this paper). Thus this paper is a welcome and timely addition to the existing toolkit of PBRL techniques.
- **[S2]** I found the exposition clear, and the motivation for various parts of the algorithm was solid. The ablations were also adequate at demonstrating the need for the most complex parts of this algorithm (the regularizers/auxiliary losses that encourage low average policy divergence & help the algorithm learn from unlabeled demonstrations, respectively).
- **[S3]** Empirical results are strong, especially for Adroit Pen and Kitchen. The experiments section also adequately explains why these specific benchmark environments were chosen over others in the D4RL suite, which is appreciated.

**Weaknesses:**

My main concern with this paper is that the baselines might be too weak, especially given that the method looks fairly similar to BC. It would also be nice to have more analysis of the policy deviation-based loss function. In more detail:

- **[W1]** The results in Table 1 and Table 2 (Gym and Kitchen only) seem relatively weak. Table 1 of the [RvS paper](https://arxiv.org/pdf/2112.10751.pdf) shows a method that is essentially BC (RvS) getting comparable or better performance to DPPO on most of these tasks (see RvS-R column and RvS-G column—the former is reward-conditioned and the latter is goal-conditioned). Are these results applicable in this setting, and if so, why are CQL and IQL used instead of these stronger and simpler baselines?

  The regularizer in Equation (4) seems to have the effect of encouraging the policy to have lower average divergence from the labeled trajectories in general, which is vaguely BC-like, and so I think it’s particularly important to separate this method from BC-style methods (filtered BC is going to be one of the leading competitors to any PBRL method like this).
- **[W2]** The loss function in Equation (2) is a significant departure from the traditional Boltzmann PBRL loss. It seems to be saying that demonstrators prefer trajectories that deviate less, on average, from the optimal policy, or something of that nature. The paper doesn’t spend much time exploring the implications of this change, or the degree to which the new algorithm is or isn’t equivalent to traditional PBRL. Having more explanation of this would be helpful—when is this the “right thing” to do? Are there any situations in which it’s obviously the wrong thing to do? (due to how the labeller might be judging trajectory fragments) etc.

On the whole I’d be fairly comfortable accepting this paper if the concern about baselines is addressed. I’m marking as a weak accept pending the response to the points above, but I’m leaning towards acceptance overall.

**Questions:**

In addition to responding to the weaknesses above, it would be helpful if the authors could answer these minor questions:

- **[Q1]** Is Figure 2 illustrating reward or return? Also, what dataset is it using? I would not expect step-level rewards to be especially correlated given how many equivalent reward functions there are for Hopper (the goal is super simple—just run forward!), but I would expect return to be correlated on any sufficiently diverse dataset.
- **[Q2]** What do the ranges and error bars in the tables represent? (std dev, std error, CIs, min/max, etc.)

**Limitations:**

**[L1]** Depending on the response to W2 above, it might be appropriate to expand the limitations with more discussion of when equation (2) is the right loss function to pursue. I expect this could tie in with the existing remarks about label noise—being more explicit about the assumptions behind equation (2) will probably reveal what kinds of label noise it can or can’t handle well.

---

> ### Author Rebuttal · Authors · 2023-08-09
>
> We express our gratitude to the reviewer for their positive remarks and insightful feedback. We would like to address your comments below.
>
> **1. ... examples include LLMs and real robots—although neither are used in this paper ...**
>
> We concur with the reviewer that domains such as LLMs and real robots would specifically benefit from PbRL. Accordingly, we have added an LLM RLHF experiment with DPPO in Appendix Section C of our paper. This experiment has yielded encouraging outcomes, indicating that DPPO can effectively supplant the reward-modeling plus PPO method typically employed in RLHF.
>
> **2. Comparison with BC-based offline RL algorithms.**
>
> Thank you for the suggestion. We first note that RvS is not directly applicable to the PbRL context as RvS requires explicit reward or goal information, which is not provided in conventional PbRL scenarios. Additionally, our initial selection of IQL (and CQL) was based on the value-based methods adopted by the baseline PT. That said, we present the results of BC-based offline algorithms in the common response 4. These results demonstrate that our method markedly surpasses the BC-based techniques.
>
> **3. Clarification on Equation (2).**
>
> Thank you for the valuable discussion on our learning objective. We would like to clarify that we continue to employ the Boltzmann PbRL loss as shown in Equation (5) (Preference Correctness term). The preference model optimizes the Boltzmann PbRL loss (supplemented with a smoothness regularizing term) with regard to the given preference labels. The core distinction between the reward-modeling baselines and our approach is that we do not model the preference as a weighted sum of rewards. Rather, we model the preference directly. Equation (2) represents the policy optimization objective used after earning the preference model.
>
> **4. Clarification on Figure 2.**
>
> Figure 2 illustrates the predicted reward vs. the true reward on the hopper-medium-replay dataset. Aligning with the reviewer's observation that the hopper environment is relatively simplistic and may be represented by numerous alternative reward functions, we performed a similar experiment on the much more complex pen-human dataset. The results are presented in Figure 1 (please check out the pdf file in the common response). The new results reveal that the baseline reward model remains ineffective at accurately capturing the true reward structure.
>
>
> **5. What do the ranges and error bars in the tables represent?**
>
> The ranges and the error bars indicate the standard deviation regarding the random seeds (Table 1, 2, Figure 6, 7) or the environments (Figure 8). We will make this clearer in the updated manuscript.

---

> > ### Comment · Reviewer_CbsJ · 2023-08-18
> > **Response**
> >
> > ## Overall opinion
> >
> > The authors resolved my main concern (the baselines) and I've increased my score accordingly. Detailed comments are given below.
> >
> > ## Detailed response
> >
> > > Accordingly, we have added an LLM RLHF experiment with DPPO in Appendix Section C of our paper. This experiment has yielded encouraging outcomes, indicating that DPPO can effectively supplant the reward-modeling plus PPO method typically employed in RLHF.
> >
> > Nice! Is this meant to be visible in the global/top-level rebuttal, or are you just saying you'll add it to the camera-ready?
> >
> > > 2. Comparison with BC-based offline RL algorithms.
> >
> > Thanks for running these. This resolves my main concern with the paper.
> >
> > > The core distinction between the reward-modeling baselines and our approach is that we do not model the preference as a weighted sum of rewards
> >
> > To be clear, the thing I was after was some discussion of whether this new assumption is more or less appropriate for real data than the Boltzmann rationality assumption. See [this blog post](https://iliad.stanford.edu/blog/2020/03/19/when-humans-arent-optimal-robots-that-collaborate-with-risk-aware-humans/) as an example of this kind of analysis: they compare the standard Boltzmann model (search for "The Rational Model") with some alternative model ("Our Risk-Aware Model") and argue that their model is a better fit for actual human demonstrators by making reference to psych/econ literature. I don't think this level of analysis is necessary for publication, but I do think it's at least worth acknowledging that this paper that the proposed method is making different assumptions about the demonstrator to traditional Boltzmann rationality.
> >
> > > hopper environment is relatively simplistic and may be represented by numerous alternative reward functions
> >
> > Complexity is not the problem, the problem is using reward (which is assigned at each transition) rather than return (which is the sum of all rewards accrued over a trajectory). Two reward functions could differ by [a potential-based shaping term](https://people.eecs.berkeley.edu/~pabbeel/cs287-fa09/readings/NgHaradaRussell-shaping-ICML1999.pdf) but still yield exactly the same optimal policy set on all MDPs. For a specific MDP, there could be even more transformations that you do and still get a semantically equivalent reward function. An example: if you want your robot to run forward, you could give it a reward proportional to its displacement only at the end of the trajectory, or you could give it a reward at each step equal to its velocity. Both will yield the same optimal policy set. Comparing returns is more robust to this problem.
> >
> > > The ranges and the error bars indicate the standard deviation regarding the random seeds (Table 1, 2, Figure 6, 7) or the environments (Figure 8). We will make this clearer in the updated manuscript.
> >
> > Thanks for the clarification.

---

> > > ### Author Response · Authors · 2023-08-21
> > >
> > > Thank you for the response!
> > >
> > > We are glad that our initial rebuttal has resolved your main concern. We would like to address your additional concerns below:
> > >
> > > **1. Clarification on the LLM experiments**
> > >
> > > Actually, the LLM experiment results are already included in our Appendix (Section C), so please check it out! Sadly, as the experiments were only finished a few days after the main paper submission deadline, we were not able to present the results in the main paper or direct the readers to check out the results in the appendix.
> > >
> > > **2. Discussion regarding the Boltzmann rationality assumption**
> > >
> > > Thank you for the detailed clarification and the valuable article provided. To discuss this matter in detail, we argue that our new assumption, which models the preference directly, aligns more closely with human preference than the conventional assumption, which models the preference as a sum of the underlying rewards. This is rooted in the observation that human decision-making tends to be non-Markovian. For some tasks, it can be challenging to design a state space rich enough to make the reward Markovian, hence making the problem formulation non-Markovian due to practical issues [1]. Also, from a psychological perspective, the human experience of a trajectory is likely to be temporally extended, rather than being independent between each timestep [2, 3]. We will include this discussion in the updated manuscript.
> > >
> > > **3. On measuring rewards instead of returns in Figure 2**
> > >
> > > Thank you for the detailed clarification. We agree with your argument that comparing the rewards may not provide the most price measure as alternative reward functions might be present. In the updated manuscript, we will re-design our Figure 2 experiment by comparing returns.
> > >
> > > Once again, we express our gratitude to the reviewer for the insightful feedback during the rebuttal period.
> > >
> > > [1] Bacchus et al., Rewarding Behaviors, National Conference on Artificial Intelligence 1996.
> > >
> > > [2] Early et al., Non-markovian reward modeling from trajectory labels via interpretable multiple instance learning, NeurIPS 2022.
> > >
> > > [3] Christiano et al., Deep reinforcement learning from human preferences, NeurIPS 2017.

---

### Author Rebuttal · Authors · 2023-08-09

We are grateful to the reviewers for their supportive and helpful feedback. We plan to address some of the shared concerns raised by multiple reviewers in this section, while individual responses will deal with more specific issues. Moreover, we'd like to emphasize that section C of our appendix showcases an LLM RLHF experiment with DPPO. This experiment reveals an encouraging outcome, demonstrating that our algorithm can successfully fine-tune LLMs, replacing the traditionally employed PPO algorithm.

**1. Clarification on the difference between the preference model from DPPO and the reward model from PT (PreferenceTransformer).**

We clarify that our preference model does not calculate the preference score (model output) as a weighted sum of some scalar value for each timestamp. Concretely, we design our transformer to output a hidden embedding for each timestamp, compute the mean of these hidden embeddings, and then pass the mean embedding through an MLP layer to generate a scalar preference score. Note that this kind of structure is common for tasks such as text classification. The only major resemblance between our preference model and the reward model from PT is that we use GPT-2 as the foundational transformer architecture to ensure a fair comparison with the baseline regarding model capacity.

**2. Some related works are missing.**

We thank the reviewers for highlighting some important works aligning with our method. In particular, [1] bears similarity to our work as it leverages the preference information to directly optimize the policy using the idea of trajectory distance. However, a limitation of [1] is that it requires the two trajectories being compared to (1) start from the same initial state and (2) roll out from the current policy being trained, which makes it challenging to utilize extensive pre-collected offline data. We will also incorporate other related works recommended by the reviewers in our revised manuscript, thereby enriching the related works section to adequately address the relevant literature.

**3. Applying the smoothness regularizer to the baseline methods.**

In recognition of the fact that our novel smoothness regularizer can be integrated into any reward-modeling baselines, we assess how its application to the baselines would influence the overall performance.

|  | PT+CQL | PT+CQL+reg | PT+IQL | PT+IQL+reg | DPPO (Ours) |
|---|---|---|---|---|---|
| pen-human | 31.6 $\pm$ 3.3 | 18.3 $\pm$ 17.2 | 53.0 $\pm$ 31.7 | 53.7 $\pm$ 42.3 | **76.3 $\pm$ 14.4** |
| pen-cloned | 18.3 $\pm$ 10.6 | 32.7 $\pm$ 11.2 | 42.9 $\pm$ 24.4 | 49.8 $\pm$ 32.2 | **75.1 $\pm$ 7.7** |
| Average | 25.0 | 25.5 | 48.0 | 51.8 | **75.7** |
| kitchen-mixed | 12.3 $\pm$ 7.7 | 12.0 $\pm$ 5.0 | 48.0 $\pm$ 11.9 | 49.4 $\pm$ 5.2 | **52.5 $\pm$ 3.1** |
| kitchen-partial | 14.1 $\pm$ 13.0 | 11.4 $\pm$ 11.2 | 40.2 $\pm$ 12.3 | **49.4 $\pm$ 5.2** | **49.4 $\pm$ 5.7** |
| Average | 13.2 | 11.7 | 44.1 | 49.4 | **51.0** |

Although our smoothness regularizer does enhance the baseline methods' performance, this improved performance still lags behind our proposed method. This outcome suggests that the process of directly optimizing the policy through the use of preference information plays a critical role in the overall performance.

**4. Comparison with BC-based offline RL algorithms.**

Here, we compare our approach with certain BC-based offline algorithms, such as RvS [2] and filtered BC. Drawing from PT+IQL or PT+CQL, we modify the RvS algorithm to utilize the reward model from PT for predicting rewards for each state-action pair. These predicted rewards are then used to run RvS, referred to as PT+RvS. Along with PT+RvS, we also implement PT+filtered BC by using the predicted rewards from PT to rank the full trajectories and perform BC on the top 10% of the trajectories.

|  | PT+RvS | PT+filtered BC | DPPO (Ours) |
|---|---|---|---|
| pen-human | -1.8 $\pm$ 0.5 | 19.4 $\pm$ 6.7 | **76.3 $\pm$ 14.4** |
| pen-cloned | -2.2 $\pm$ 0.3 | 37.4 $\pm$ 7.5 | **75.1 $\pm$ 7.7** |
| Average | -2.0 | 28.4 | **75.7** |
| kitchen-mixed | 27.5 $\pm$ 9.4 | 40.9 $\pm$ 9.0 | **52.5 $\pm$ 3.1** |
| kitchen-partial | 26.0 $\pm$ 4.9 | **53.4 $\pm$ 9.0** | **49.4 $\pm$ 5.7** |
| Average | 26.8 | 47.2 | **51.0** |

The results show that DPPO outperforms the BC-based baselines on all tasks, with the performance gap being especially large on the challenging pen tasks.

[1] Wilson et al., A Bayesian approach for policy learning from trajectory preference queries, NeurIPS 2012.

[2] Emmons et al., What is Essential for Offline RL via Supervised Learning?, ICLR 2022.

---

### Decision · Program_Chairs · 2023-09-21

**Decision:**

Accept (poster)

**Comment:**

The meta-reviewer has reviewed the paper, the reviews, the responses, and agrees with the majority of the reviewers that this paper meets the NeurIPS standard.
There is a large number of Preference-based RL papers coming out recently. This paper is a borderline accept in terms of novelty and significance.